



# The Baltic Sea model inter-comparison project BMIP – a platform for model development, evaluation, and uncertainty assessment

Matthias Gröger[1], Manja Placke[2], H.E. Markus Meier[1,3], Florian Börgel[1], Sandra-Esther Brunnabend[3], Cyril Dutheil[1], Ulf Gräwe[1], Magnus Hieronymus[3], Thomas Neumann[1], Hagen Radtke[1], Semjon
Schimanke[3], Jian Su[4], Germo Väli[5]

[1]Department of Physical Oceanography and Instrumentation, Leibniz Institute for Baltic Sea Research Warnemünde, Rostock, 18119, Germany
[2]IT Department, Leibniz Institute for Baltic Sea Research Warnemünde, Rostock, 18119, Germany
[3]Research and Development Department, Swedish Meteorological and Hydrological Institute, Norrköping, 60176, Sweden
[4]Danish Meteorological Institute, Lyngbyvej 100, DK-2100 Copenhagen, Denmark
[5]Department of Marine Systems, Tallinn University of Technology, Tallinn, Estonia

Correspondence to: Matthias Gröger (matthias.groeger@io-warnemuende.de)

**Abstract**.

While advanced computational capabilities have enabled the development of complex ocean general circulation models (OGCM) for marginal seas, systematic comparisons of regional ocean models and their setups are still rare. The Baltic Sea model inter-comparison project (BMIP), introduced herein, was therefore established as a platform for the scientific analysis and systematic comparison of Baltic Sea models. The inclusion of a physically consistent regional reanalysis data set for the period 1961–2018 allows standardized meteorological forcing and river runoff. Protocols to harmonize model outputs and
analyses are provided as well.

An analysis of six simulations performed with four regional OGCMs differing in their resolution, grid coordinates, and numerical methods was carried out to explore inter-model differences despite harmonized forcing. Uncertainties in modeled surface temperatures were shown to be larger at extreme than at moderate temperatures. In addition, a roughly linear increase
in the temperature spread with increasing water depth was determined and indicated larger uncertainties in the near-bottom layer. On the seasonal scale, the model spread was larger in summer than in winter, likely due to differences in the models' thermocline dynamics. In winter, stronger air-sea heat fluxes and vigorous convective and wind mixing reduced the inter-model spread. Uncertainties were likewise reduced near the coasts, where the impact of meteorological forcing was stronger. The uncertainties were highest in the Bothnian Sea and Bothnian Bay, attributable to the differences between the models in
the seasonal cycles of sea ice triggered by the ice-albedo feedback. However, despite the large spreads in the mean climatologies, high inter-annual correlations between the sea surface temperatures (SSTs) of all models and data derived from a satellite product were determined. The exceptions were the Bothnian Sea and Bothnian Bay, where the correlation dropped significantly, likely related to the effect of sea ice on air-sea heat exchange.

Marine heat waves (MHWs), coastal upwelling, and stratification were also assessed. In all models, MHWs were more frequent in shallow areas and in regions with seasonal ice cover. An increase in the frequency (regionally varying between ~50 and 250%) and duration (50–150%) of MHWs during the last three decades in all models was found as well. The uncertainties were highest in the Bothnian Bay, likely due to the different trends in sea-ice presence. All but one of the analyzed models overestimated upwelling frequencies along the Swedish coast, the Gulf of Finland, and around Gotland
while they underestimated upwelling in the Gulf of Riga. The onset and seasonal cycle of thermal stratification likewise differed among the models. Compared to observation-based estimates, in all models the thermocline in early spring was too deep whereas a good match was obtained in June, when the thermocline intensifies.





## 1 Introduction

Coordinated model experiments are common practice in global ocean model modeling, as exemplified by the ocean model inter-comparison project (OMIP, Griffies et al., 2016) which seeks to identify systematic model biases and to address inter-model differences between participating models. However, parallel efforts in modeling regional seas are still rare and have mostly focused on wider open ocean regions, such as the Arctic (e.g. the Arctic Ocean Model Comparison Project, https://web.whoi.edu/famos/) and the North Atlantic (Barnard et al., 1997). Shelf seas have yet to be systematically studied

despite their high economic importance. For the shallow Baltic Sea and North Sea, only a few, non-systematic studies have included inter-model comparisons (e.g., Myrberg et al., 2010; Eilola et al. 2011; Placke et al. 2018; Pätsch et al. 2017). Hence, in the following we introduce the Baltic Sea Model Intercomparison Project (BMIP). The Baltic Sea is an estuarine sea on the NW European shelf and is an important factor in the economies of nine European countries (Russia, Finland, Estonia, Latvia, Lithuania, Poland, Germany, Denmark, and Sweden). However, unlike other marginal seas the Baltic Sea

has become highly eutrophic, due to agricultural and industrial inputs from the hydrological catchment area. Furthermore, the impact of climate warming is expected to be high (e.g. Meier et al., 2018; Meier et al., 2019; Saraiva et al., 2018 Gröger et al., 2019, Dieterich et al., 2019; Meier et al., 2021, Meier et al., 2022; Gröger et al., 2021a; Gröger et al., 2021b, Gröger et al., 2022; Wahlström et al., 2020; Wahlström et al., 2022).

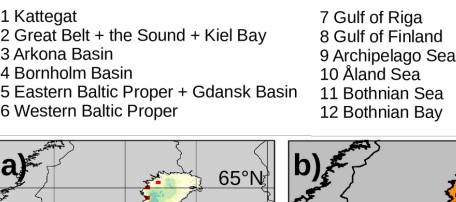

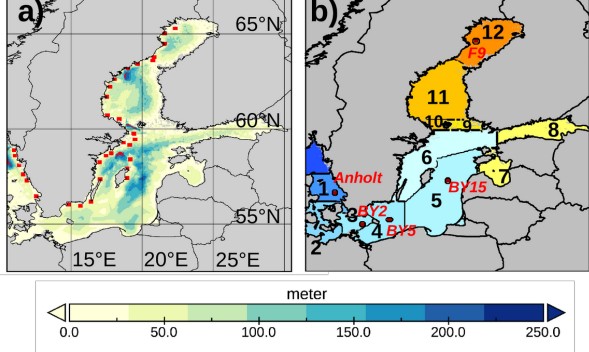

*Figure 1: a) Bathymetry of the Baltic Sea. Red boxes indicate the positions of the Swedish stations used in wind and temperature analyses (see Suppl. Mat. S1). b) Basin division for the Baltic Sea according to Meier et al. (1999). Red circles indicate stations used for model vs. data comparisons.*

The Baltic Sea is among the most complicated regions of the world ocean, given the complex bathymetry with several sub-basins (Fig. 1) and the limited water exchange between them. The estuarine character of the Baltic Sea is due to sporadic saltwater intrusions from the North Sea, which are the product of complex overflows occurring across the Great Belt and Sound region (Fig. 1) and lead to a permanent halocline between 60 and 80 meters depth (Väli et al., 2013). The long history

of oceanographic research in the Baltic Sea has resulted in numerous, very diverse models, ranging from simple box models (e.g., Knudsen, 1900; Welander, 1974), to process-oriented models (e.g., Stigebrandt, 1983; 1987; Omstedt, 1990; Omstedt


and Axell, 2003), and, later, to general circulation models (GCM). The latter include advanced methods for the vertical and horizontal discretization of partial differential equations for momentum, energy, and mass conservation at fine-resolution grids as well as for various empirical sub-grid-scale parameterizations (e.g., Meier et al., 1999; Meier, 2001; Myrberg et al.,

2010; Hordoir et al., 2019). An overview of the history of regional climate modeling for the Baltic Sea and its surrounding catchment area since the 1990s using GCMs was provided by Meier and Saraiva (2020).

A first initiative to systematically investigate physical properties of the Baltic Sea using multiple models focused on the Gulf of Finland (Myrberg et al., 2010). The authors compared six different 3D hydrodynamical models which were driven by the

same atmospheric forcing, initial conditions, as well as the same model grid with a resolution of 4 x 2 minutes (Myrberg et al. 2010). The study identified common difficulties in representing the mixed layer dynamics resulting in biases in vertical temperature and salinity profiles. The authors emphasized the need for higher resolution and more advanced mixing schemes as well as accurate inputs of river discharge. However, the simulations comprised only the summer-autumn 1996 and thus, did not allow to assess the long term climate variability


The as yet largest, but uncoordinated ensemble of scenario simulations for the Baltic Sea was analyzed by Meier et al. (2018) and the uncertainties of these projections were discussed in a subsequent publication (Meier et al. 2019a). As the model simulations during the historical period differed from observations, and with mismatches between ocean models attributed to differences in atmospheric forcing, it was concluded that model performance must be rigorously assessed to improve future

projections and to reduce the spread among models.

Accordingly, Placke et al. (2018) examined water-mass circulation in different hydrodynamical models for the 30-year period covering 1970–1999 and compared the results with reanalysis data. They found that a substantial portion of the inter-model differences could be explained simply by the different wind forcings and by the riverine freshwater inputs used to

force the models. In addition, they showed that, compared with observations, newer ocean circulation models did not always perform better than the first Baltic Sea models, which were developed 20 years ago.

During recent decades more powerful computational facilities have allowed the development of increasingly complex numerical methods (e.g., horizontal advection schemes, adaptive vertical coordinates, unstructured grids). In addition

advanced schemes for sub grid-scale parameterizations (e.g. horizontal and vertical turbulence) were developed. In parallel, the amount of available forcing data describing river discharge and the atmospheric boundary layer has also increased. However, comparisons of the internal process formulations of these different models require a harmonization of the experimental design (spin-up, initialization, open lateral boundary conditions, atmospheric forcing, river discharge). Moreover, despite advancements in model development, no new attempts have been made to systematically compare and

validate Baltic Sea models since the studies of Myrberg et al. (2010) and Eilola et al. (2011).

The BMIP can close this gap by providing a coordinated framework for experimental design, model output, and analyses of model results. Among the aims of the project are the development and provision of driving data for the most important forcings of Baltic Sea models, i.e., atmospheric boundary data, river discharge, and lateral boundary data. Furthermore, the

BMIP includes recommendations for model initialization and spin-up. The overall goal of this community effort is to improve the quality of Baltic Sea models, especially for climate variability, and, in turn, climate impact research.





Thus, in this first BMIP paper, the focus is on the models used for climate simulations, i.e., models that can be integrated over several decades with reasonable resources. However, the BMIP also considers models that were developed for
operational short-term marine forecasts (e.g., sea level, sea ice), such as the HIROMB-BOOS ocean circulation model (HBM) from the Danish Meteorological Institute (Berg and Poulsen, 2012). Over longer time scales, the performance of these models can be expected to deteriorate, when they are driven with data assimilation but evolve freely. Consequently, these models have rarely been validated with respect to their long-term performance, such as in multi-decadal transient simulations. Finally, the BMIP also includes model setups with horizontal resolutions in the range of a few tens of meters to
~200 meters, as they allow the resolution of sub-mesocale dynamics and mesoscale eddy fields (Väli et al. 2017; Väli et al., 2018; Zhurbas et al., 2019; Onken et al., 2020).

The paper is organized as follows: Section 2 provides a description of the forcing data sets to be used in the BMIP and outlines the protocol to set up a BMIP run. Section 3 assesses the results of six hindcast simulations from four different
model platforms. Section 4 compares topical case studies for marine heat waves (MHWs) coastal upwellings, and water stratification. Section 5 discusses aspects of ultra-high resolution modeling (~250 m) within BMIP. A summary and the main conclusions constitute Section 6.

## 2. Methods

**2. 1. Forcing data**

**Runoff**

A homogeneous data set describing freshwater input to the Baltic Sea was produced within the BMIP project (Fig. 2) with the aim of forcing each of the Baltic Sea models with identical runoff. The new data set is based on the runoff hindcast obtained with the pan-European hydrology model E-HYPE (Lindström et al. 2010) and forced by meteorological ERA-
interim data (Dee et al. 2011) that were downscaled using the regional atmosphere model RCA3 (Samuelsson et al. 2011) for the period 1979–2012. For the period 2012–2018, an E-HYPE model forecast product (Donnelly et al. 2016) was used. For the early period 1961–1978, climatological runoff data from 1979 to 2008 had to be used but they were scaled by the annual mean values for the period 1961–1978 reported by Bergström and Carlsson (1994). For the Neva River, the largest river in the eastern Gulf of Finland, daily observations for 1961–2016 were provided by the Russian State Hydrological Institute
(Sergei Zhuravlev, personal comm.). For detailed information on the runoff dataset the reader is referred to Väli et al (2019).





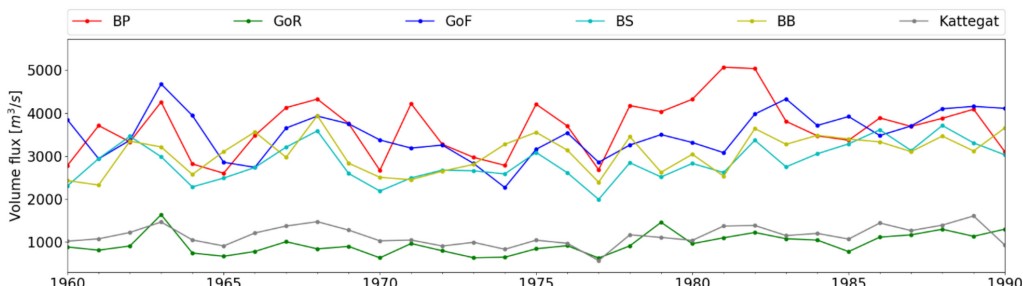

*Figure 2: Freshwater input to different sub-basins of the Baltic Sea from the BMIP runoff forcing for the years 1960-1990. Figure adopted from Väli et al (2019). BP: Baltic proper, GoR: Gulf of Riga, GoF: Gulf of Finland, BS: Bothnian Sea, BB: Bothnian Bay.*

**The European Regional Reanalysis UERRA (version 1.0)**

The regional reanalysis data set UERRA-HARMONIE was chosen as the atmospheric forcing for the present OMIP as it

provides a physically consistent data set over almost 60 years and thus fit the requirements for transient multi-decadal simulations. The UERRA-HARMONIE reanalysis system was developed within the FP7 project UERRA (Uncertainties in Ensembles of Regional Re-Analyses, http://www.uerra.eu/). The data set was initially produced in the UERRA project and then carried over to the Copernicus Climate Change Service (C3S, https://climate.copernicus.eu/copernicus-regional-reanalysis-europe). UERRA-HARMONIE is a long-term, high-quality, high-resolution regional reanalysis that includes

many essential climate variables. Data on air temperature, pressure, humidity, wind speed and direction, cloud cover, precipitation, albedo, surface heat fluxes, and radiation fluxes are available for the period January 1961 through July 2019, which is long enough for climatological analyses. UERRA-HARMONIE has a horizontal resolution of 11 km, with analyses carried out at 00 UTC, 06 UTC, 12 UTC, and 18 UTC; data from the forecast model with hourly resolution are also provided. UERRA-HARMONIE is available via Copernicus Climate Data Store (CDS,

https://cds.climate.copernicus.eu/#!/home). The parameters needed in the forcing of the ocean models belong to the category single-level data and can be directly accessed from: https://cds.climate.copernicus.eu/cdsapp#!/dataset/reanalysis-uerra-europe-single-levels?tab=overview.

The data are freely available upon registration and acceptance of the license. Within the Copernicus User Learning Service

(ULS) GitHub (https://github.com/UserLearningServices-C3S/regionalreanalysis-UERRA), an example of data access and preparation is provided using the NEMO-Nordic model (Hordoir et al., 2019). Shortcomings in UERRA-HARMONIE, e.g., for precipitation or cloudiness, are explained in the instruction file from the BMIP website (https://www.baltic.earth/working_groups/model_intercomparison/index.php.en), which also offers solutions on how to deal with those parameters. Both a brief assessment of the atmospheric data with respect to observations and the ERA5 reanalysis

data are available in Suppl. Mat. S1.

**2.2 Ocean models**

Six configurations based on four different model platforms (GETM, MOM, HBM, NEMO) were assessed in this study, with

a focus on the models' capability to describe long-term climatologies and dynamics. While the GETM, MOM, and NEMO





were designed for free long-term integrations of multiple decades, the HBM is primarily used for short-term operational services and was thus designed to mainly operate with data assimilation techniques. In the BMIP, it was run for the first time in free mode.

Table 1 provides information on the model setups assessed in this study. The GETM_1nm and GETM_2nm domain is limited to the southern Kattegat, while that of the two MOM domains also include parts of the Skagerrak. Both the NEMO_2nm and the HBM_3nm encompass the North Sea, for which they also consider tidal forcing. The horizontal resolution of these models is between 1 and 3 nautical miles (nm). GETM_hires was integrated only for a few months, as it is too expensive for multi-decadal simulations. The NEMO_2nm model incorporates a multi-class dynamical ice model

while the other models include simpler Hibler-type models. The model setups vary strongly in their vertical discretization. Thus, while the GETM uses 60 vertical adaptive terrains following s-coordinates (Hofmeister et al. 2010), the other models have z*-coordinates that at every time step are re-scaled to the actual sea surface height. Surface layer thicknesses ranges from 0.25 m (GETM) to 8 m (HBM). All models use the radiative fluxes (downward long-wave and downward short-wave) provided by the BMIP forcing but differ in their calculation of momentum flux (wind stress), sensible and latent surface heat

fluxes as well as in their upward long-wave radiation, which were estimated using different bulk formulas from the other surface fields provided by the BMIP. A short description of each model along with further references regarding details of the respective physics can be found in Suppl. Mat. S2.

| | Horizontal resolution | Vertical levels /first layer thickness | Domain | tides | Sea ice | Bulk formula (non-radiative air – sea fluxes) |
|---|---|---|---|---|---|---|
| GETM_1nm | 1 nm | 60 /0.25m s-levels, vertically adaptive | BalticSea+ southern Kattegat | – | Hibler type, Winton (2000) | Kara et al., 2005 |
| GETM_2nm | 2 nm | 60 /0.25m s-levels, vertically adaptive | BalticSea+ southern Kattegat | – | Hibler type, Winton (2000) | Kara et al., 2005 |
| GETM_hihres | 250 m | 60 / 0.5m vertically adaptive | BalticSea+ southern Kattegat | – | Hibler type, Winton (2000) | Kara et al., 2005 |
| MOM_1nm | 1 nm | 152 z* levels / 0.5m | Baltic proper | – | Hibler type, Hunke, E. C. and Dukowicz, J. K. (1997), Winton (2000), | Based on Large and Yeager, 2004 |
| MOM_3nm | 3 nm | 152 z* levels / 0.5m | Baltic Sea+eastern Skagerrak | – | Hibler type, Hunke, E. C. and Dukowicz, J. K. (1997), Winton (2000), | Based on Large and Yeager, 2004, |
| HBM_3nm | 3 nm | 50 / 8m | Baltic Sea+North Sea | 17 constituents | Hibler type, Kleine and Skylar, (1995) | Andree et al. (2021) |
| NEMO_2nm | 2 nm | 56 z* -level / 3m | Baltic Sea+North Sea | 12 constituents | Dynamic ice model with multiple ice classes), Vancoppenolle et al., (2009) | Based on Large and Yeager, 2004 |

*Table 1: Overview of the models used in this study (nm=nautical miles).*



### 2. 3 The BMIP protocol version 1.0

The BMIP was invoked to establish atmospheric and hydrological forcing data and to develop best practices in the set up of climate simulations for the Baltic Sea. Discussions within the international project group have addressed the ability of state-of-the-art ocean general circulation models to sufficiently represent climate-relevant ocean processes, the required grid resolution, improving parameterizations specific for the Baltic Sea's physics (i.e., the sea's variable topography, estuarine circulation due to excessive freshwater input, and the impact of tides), and (in a second phase) marine biogeochemistry. The

aim of BMIP is to improve the performance of Baltic Sea simulations, for both past and future climates, and to foster international scientific collaboration on ocean climate model development and setup.

The forcing data and ocean model diagnostics provided by the BMIP are appropriate for the Baltic Sea but the methods are nevertheless likely to be applicable to other marginal seas worldwide. In particular, the BMIP aims to establish a framework

for:

- ocean model and sea ice model development and validation
- comparisons of model results with data products, followed by an understanding of the reasons of the differences between them

- investigating physical and (later) biogeochemical processes ranging from sub-mesoscale dynamics to multi-decadal (climate) variations

A BMIP simulation can be set up by following the instructions on the project's web portal (https://www.baltic-earth.eu/working_groups/model_intercomparison/index.php.en). Data on 2-m air temperature [K],

precipitation [kgm$^{-2}$], snowfall [kgm$^{-2}$], downward long-wave radiation [Jm$^{-2}$], downward short-wave radiation [Jm$^{-2}$], sea level pressure [Pa], surface humidity [%], and 10-m wind components [ms$^{-1}$] can also be downloaded from the web site. Data on cloudiness fields are not provided because they were corrupted during the production of the UERRA data set. Thus, for models that calculate longwave radiation from cloudiness, the use of ecoastDat-2 data is recommended (Geyer et al., 2014). No data on initial fields are provided. Since the Baltic Sea has low overturning rates, a 23-year long spin-up integration,

from 1961 to 2004, is recommended to reduce strong model drifts in the first decade. As major Baltic inflows (MBIs; Matthäus and Frank, 1992; Schinke and Matthäus, 1998) can cause deep-water properties and thus the stability of the static water column to change abruptly, the production run starting from 1961 should be launched with the initial fields from July 1, 2004, taken from the spin-up run.

Due to the large horizontal and vertical temperature and salinity ranges that characterize the Baltic Sea, the horizontal and vertical resolution should be high. Thus, the horizontal resolution is ideally set to 2 nm or higher but it should not be coarser than 10 km, to allow reasonable comparisons with other models and with observation data. For z-level or z*-level (Levier et al. 2007; Campin et al., 2008) coordinates, the vertical grid spacing should be at least 2 m in order to reasonably cover the strong temperature and salinity gradients that occur across the summer thermocline and the perennial halocline.


Model output and diagnostics can be derived from the BMIP web site. For halocline, thermocline, and pycnocline diagnostics, separate algorithms are provided (https://owncloud.io-warnemuende.de/index.php/s/LVZbDvSvcTnECpb). For these parameters at least daily temperature and salinity data are recommended. Detailed instructions on how to set up a



BMIP hindcast simulation are available at the BMIP project site (https://www.baltic-earth.eu/imperia/md/assets/baltic_earth/
baltic_earth/baltic_earth/baltic_earth/bmip_instructions.pdf).

The objective of the assessment presented below was to identify systematic differences between models from Denmark, Estonia, Germany, and Sweden, despite the common forcing. A comprehensive validation for each model is beyond the scope of this study.


### 2.4. Analysis of heat waves, coastal upwelling, and water column stratification

**Heat waves**

MHWs were analyzed following Hobday et al. (2018). For every grid cell, first, the multi-year daily mean SST climatology
was calculated over the reference period 1970–1999. The 90[th] percentile SST was then calculated in the same way. The daily mean climatology and the percentile were calculated for each calendar day within a 11-day window centered around the respective day. This was necessary to ensure robust estimations of the mean values and of percentile values. Heat waves were thereafter classified according to multiples of the difference between the mean climatology and the percentile. Hence, if the simulated daily SST at a given day exceeded the mean SST climatology for that day by a factor of >1, the day was
classified as a moderate MHW. Excess factors of 2, 3, and 4 denoted strong (class II), severe (class III), and extreme (class IV) MHWs. Finally, for each of the classes the total area occupied by the respective class was calculated from the daily SST series.

**Coastal Upwelling**

The upwelling analyses were based on the daily averaged SSTs of four hindcast simulations: HBM, NEMO, and the low-resolution versions of GETM and MOM (i.e., GETM_2nm, and MOM_3nm). For comparison, SST data from the AVHRR satellite at 1-km resolution were used. The satellite SST data were manually post-processed by the Bundesamt für Seeschifffahrt und Hydrographie (BSH) in order to unmask upwelling. This was necessary because the cloud detection algorithm may identify sharp gradients at the edge of the upwelling regions as clouds and thus flags these values as missing.


These datasets, covering the period 1993–2010, were re-gridded by bilinear interpolation on the coarsest grid (i.e., the HBM_3nm model) to avoid interpolation artifacts. The upwelling frequency was calculated using the method proposed in Lehmann et al. (2012), which is based on the temperature difference between the coastal SST and the surrounding water. Thus, to detect an upwelling event the temperature difference between each pixel and the zonal mean corresponding to that
pixel was calculated. An upwelling was defined as a difference lower than −2°C. Finally, a mask was applied to remove all points located beyond 28 km from the coast. As this method is based on a difference with the zonal mean, it is limited to regions where the coastline is mainly oriented along an east/west axis, as in the Gulf of Finland. Nevertheless, this automatic method was compared to a visual analysis and was shown to perform well (Lehmann et al., 2012).

**Water column stratification**

Some numerical models include an inherent option to save the depth of the mixed layer as an output variable. Comparisons of the results between models may, however, be biased by differences in how this depth is calculated. We therefore propose a common procedure to calculate the cline depths directly from the temperature and salinity fields and provide a Fortran procedure that allow this to be done either during the model run or during the postprocessing phase. The TEOS-10 equation





of state (Feistel, 2012) allows five different clines to be calculated, based on the depth of the maximum gradient between vertically adjacent model cells. Thermocline depth (td), halocline depth (sd), and pycnocline depth (rd) use a gradient of conservative temperature, absolute salinity, and density, respectively. For each of the clines, its strength, measured by the gradient (tg, sg, rg), is saved. For the other two clines, the density gradient caused by the change in one parameter alone, either the temperature difference or the salinity difference between two adjacent cells, is calculated. This allows estimations

of the thermal pycnocline depth (rtd) and the haline pycnocline depth (rsd). It further permits a direct relative comparison of the strength of thermal and haline stratification, based on comparisons of the gradients (rtg and rsg, both in kg/m$^4$). For the halocline location (sd and rsd), the 15% highest and lowest salinities in the profile were excluded to avoid the identification of thin layers of river plumes or near-bottom intrusions as the halocline.

## 3. Results


A number of modeling groups have recently started to produce BMIP model runs. Here we provide a first assessment of temperature and salinity. As the analyzed simulations differed with respect to the model initialization, results before 1970 were not interpreted. The effect of different spin-ups after 1970 on supra-halocline waters in the Baltic Sea was assumed to have been minor. For the HBM, a different runoff forcing was used. However, even with these minor deviations the model

outputs analyzed below constitute a highly harmonized data set, unlike those obtained in previous model comparisons.

### 3.1 Assessment of mean climatologies

In the following, water temperature and salinity are briefly assessed in the models. Our aim was not to provide a comprehensive validation but to demonstrate the marked differences between models despite the same forcing. For climate applications, differences in the spatial characteristics of the models should be considered (Placke et al., 2018; Gröger et al.,

2019). However, despite the long history of national and international Baltic Sea research, no long-term mean climatology product for water salinity and temperature is available that satisfyingly serves the needs of climate research (Kent et al., 2019; Zumwald et al., 2020; Hegerl et al., 2021). Therefore, for comparison we use gridded data sets of remote sensing SST data, obtained from the BSH, for the period 1990–2007. In addition a Baltic Sea reanalysis data set covering 1970–1999 (Liu et al., 2017) is provided in Suppl. Mat. S3. Both data sets are characterized by uncertainties and shortcomings, mainly arising

from limited observations in space and time. Consequently, limited observational constraints were available for the data assimilation product (lou et al., 2017). Also, no in situ data from the Baltic Sea were used in the calibration of the remotely sensed data from the BSH. The mean seasonal cycle was analyzed based on in situ data derived from the SHARK database hosted by the Swedish Meteorological and Hydrological Institute (https://sharkweb.smhi.se/hamta-data).





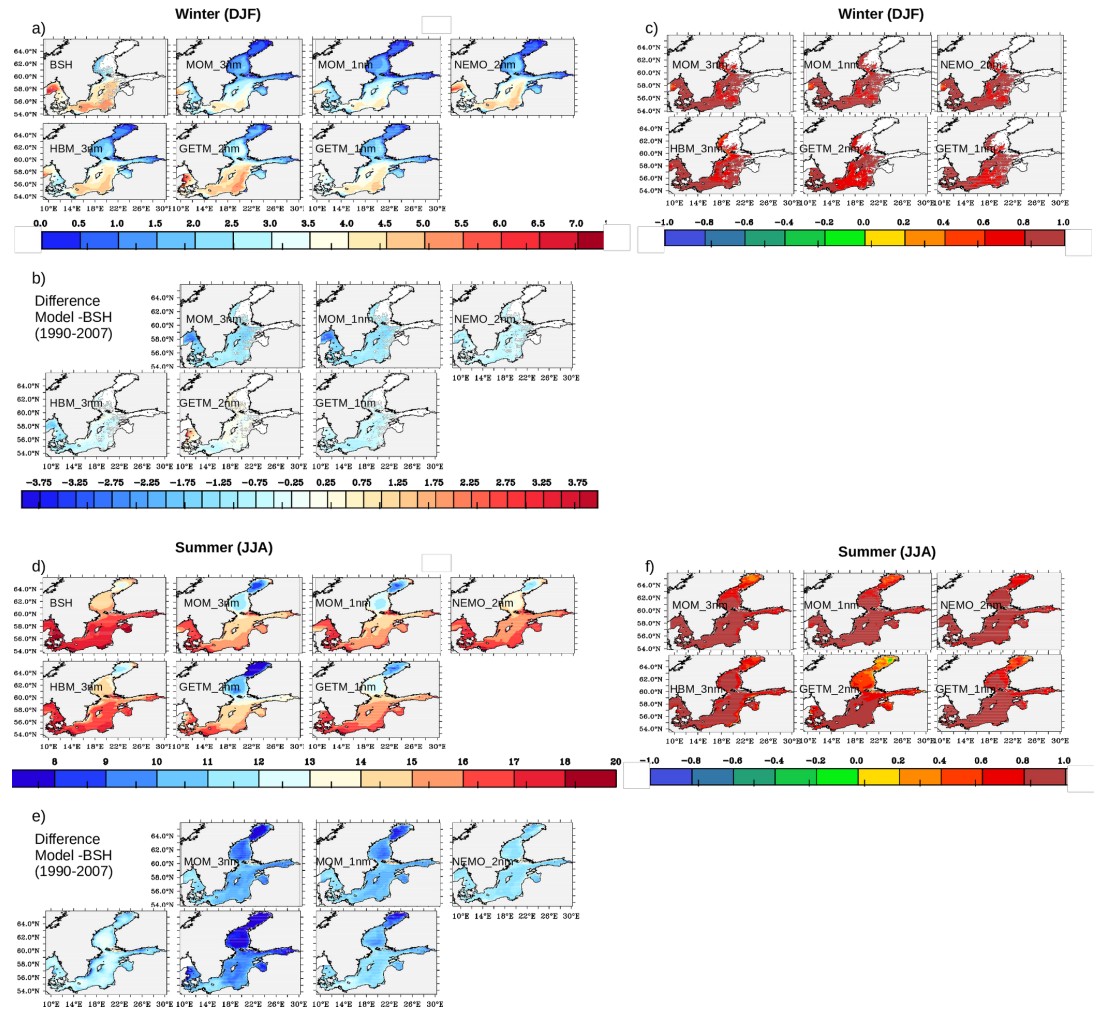

*Figure 3:a) Comparison of modeled winter SST with a satellite product from the Federal Maritime and Hydrographic Agency of Germany (BSH). b) difference between the models and the satellite product for winter. c) Inter-annual correlation of winter sea surface temperature between models and the satellite product. d-f) same as a-c) but for summer climatology. Note winter SST coverage from the satellite product is incomplete.*

In all models, winter SSTs (Fig. 3a) were lowest in the Bothnian Sea, Bothnian Bay, the Gulf of Finland, and the Gulf of Riga. In the shallower Gulf of Finland and Gulf of Riga, where the heat inventory was rather low, the SSTs adapted rapidly to the cold winter atmosphere. In the open Baltic proper, the Bornholm and Arkona basins sea ice was mostly absent such that the stronger winter winds together with convective mixing supported exchange with warmer waters from deeper layers. The satellite-based observations (Fig. 3a) revealed strong horizontal SST gradients between the cold shallow

Kattegat/Sound/Belt Sea, where the heat inventory was low and heat loss was rapid, and the deeper Skagerrak in the northeast, where vigorous cyclonic circulation and the subsequent Ekman-induced upward transport of warmer deep waters together with wind-induced deep mixing led to higher SSTs. In the models that included parts of the Skagerrak (HBM_2nm, MOM_3nm, MOM_1nm, NEMO_2nm), these gradients were also present but were generally less well pronounced than





according to the satellite product. With the exception of GETM_2nm, all models systematically simulated winter SSTs that

were lower than those in the BSH satellite data (Fig. 3b). This was also the case in comparisons between the models and the
reanalysis data set (Suppl. Mat. S3). With the exception of GETM_1nm and NEMO_2nm, the model-BSH deviations (Fig.
3b) were largest near the lateral boundaries, thus demonstrating the importance of boundary conditions in the realizations of
individual models. The high SSTs in GETM_2nm along the Danish east coast (Fig. 3a) caused strong positive anomalies in
the comparisons with the BSH climatology (Fig. 3b) and the reanalysis data (Suppl. Mat. S3).


During summer, meteorological forcing was characterized by calm winds and stronger solar radiation, which promoted an
intense thermal layering of the uppermost water column. In the open sea, air-sea coupling was affected by the presence of a
strong thermocline that reduced exchange with cooler waters from greater depths. The subsequent reduction in the effective
water column heat capacity made the SSTs more prone to variations in meteorological forcing than was the case in winter.

Similar to winter, the summer SSTs determined in the simulations were lower than those of the satellite product (Fig. 3e).
The cold deviations were considerably higher in summer than in winter and in some models exceeded −2 K. However, the
satellite product may reflect the water skin temperature (rather than the vertical mean temperature across the respective first
model layer), which was not explicitly represented in the models. Generally, the deviations between the models, the satellite
data, and the reanalysis data were much more pronounced in summer than in winter.


An important prerequisite for the use of the models in climate applications is their ability to correctly represent inter-annual
variability and to respond to long-term variations in atmospheric forcing (e.g. Gröger et al., 2015, Gröger et al., 2019).
Figure 3c shows an overall high inter-annual correlation for the winter season, with values mostly around 0.7 or higher.
Hence, despite the sometimes large discrepancies in the mean climatologies (Fig. 3e), the inter-annual variations in models

fit those in the satellite data. However, in the Bothnian Sea and Bothnian Bay in summer, the correlation values were low
and in some cases < 0.3. For the northernmost parts of the Bothnian Bay, remnant sea ice floes from the previous winter can
affect vertical mixing and affect SSTs. Thus, in these regions a realistic sea ice cover is essential.

The inter-model spread as summarized by the inter-models standard deviations (Fig. 4) is clearly higher in summer

compared to winter. However, the summer pattern also shows a significant reduction in the spread near the coasts. In these
shallow environments, no stable thermal stratification develops such that these small water bodies rapidly adapted to
identical atmospheric forcing. Notably, the models' representations of coastal upwelling along the Swedish east coast did not
increase the spread, in contrast to open sea areas, where the spread was systematically higher than in coastal regions. This
highlights the importance of the internal model dynamics that control the depth and intensity of the thermocline. The lower

inter-model spread during winter was likely related to stronger wind-induced and convective mixing, which promote a strong
heat flux out of the ocean. In areas with stable sea ice conditions, i.e., the Bothnian Bay, the eastern Gulf of Finland, and the
Gulf of Riga, the very low winter-time spread in all models could be explained by a SST roughly equal to the freezing point
temperature.

The large SST spread in the Bothnian Bay in summer may have been due to the different melting rates in the models, since
sea ice break up is highest in May/June and is followed by a warming of the surface water layer (Fig. 4).

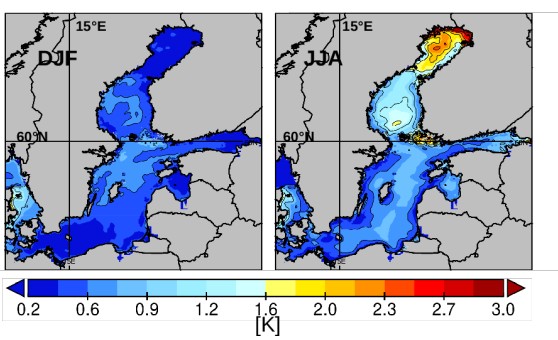

*Figure 4: Intermodel standard deviation of SST for winter (DJF) and summer (JJA). The standard deviation was calculated from the six models (MOM_3nm, MOM_1nm, GETM_2nm, GETM_1nm, HBM_3nm, and NEMO_2nm).*

### 3.3 Mean seasonal cycle

The seasonal cycle of water temperature and salinity was assessed at selected stations (Fig. 5) located at key sites along a transect that roughly followed the pathway of imported saltwater. Hence, conditions at the stations ranged from shallow waters upstream of the overflow region (Anholt, Fig. 1b), to open sea conditions in the southern Baltic (Arkona Basin BY2, Bornholm Basin BY5), to deep water conditions in the Baltic proper (east Gotland Basin, BY15) and finally to the Bothnian Bay (F9), where there is no notable halocline but seasonal ice cover has a significant effect.


The strongest seasonal cycle along the transect was determined at Anholt station  (Fig. 5), representative of the shallow water conditions in the southern Kattegat (Fig. 1b). The amplitude of the seasonal temperature cycle was most pronounced in the HBM_3nm and the two MOM models and was slightly overestimated compared to the SHARK data set. In particular, the surface to bottom temperature gradients during summer were stronger in the two MOM simulations than in the

simulations of the other models. This was in line with the likewise enhanced surface to bottom salinity gradients and suggested a generally stronger thermocline, such that mixing was underestimated in the MOM. In the NEMO_2nm, the water depth at this site was clearly shallower than in the other models such that salinities > 32 g/kg were rarely reached.

Stations BY2 and BY5 (Fig. 5) are located in the Arkona Basin and Bornholm Basin, respectively, and thus further

downstream of the overflow region. The two sites receive strong freshwater inputs from rivers while salt water is supplied by the North Sea. This results in strong vertical salinity gradients, which were most pronounced in MOM and GETM_1nm and weakest in HBM_3nm, NEMO_2nm, and GETM_2nm  (Fig. 5b). In particular, HBM underestimated salinity over the whole water column, which suggested that a potential bias in vertical mixing was not the only explanation; rather the intensities of saltwater inflows from the North Sea were likely underestimated. Furthermore, in the HBM the recommended BMIP river

runoff forcing was not applied. Runoff differences between data sets will add to the uncertainty in near-coastal salinity.

The thermal structure at BY2 and BY5 (Fig. 5a) reflects the well-studied cold intermediate layer (CIL, Liblik and Lips, 2019; Dutheil et al., 2021), the remnant of a water mass that formed during the previous winter at a depth between 20 and



60 m and became encapsulated during the subsequent warm season, along with the development of a strong thermocline. The

CIL is more pronounced at BY5, a deeper station that represents more open ocean conditions. When the storm season starts

in fall, the warmer surface waters are mixed further downward. Consequently, the surface rapidly cools while after a short

delay the intermediate water warms, finally terminating the lifetime of the CIL (Fig. 5a). This was well reproduced by all of

the studied models.

*Figure 5: Multi-year (1990-2009) mean seasonal cycle of water column temperature (a) and salinity (b) at selected monitoring sites in the Baltic Sea. BMIP models are assigned at the bottom panels (station F9). See text for further explanations. The upper left plot of each panel displays the seasonal cycle based on the SHARK data set. Note the different color scales for salinity at the Anholt and F9 stations.*





Station BY15 represents fully open sea conditions in the eastern Gotland Basin (Fig. 1b). In agreement with the
        observations, in all models the thermocline at this station was shallower than at all other considered stations (Fig. 5a). With
        further distance from the North Sea, the deep salinity becomes markedly lower than at stations BY5 and BY2. Our use of a
        common forcing data set provides the first assessment of how large BMIP models can differ due to their internal dynamics,
        such as vertical mixing or inflows. In addition to the HBM which was not designed to focus on MBIs, the GETM_2nm and
NEMO_2nm showed that salinities were lowest in the deep layer but highest in the upper layers, suggesting stronger vertical
        mixing. Stronger mixing was also reflected by the rather low vertical salinity gradients.

        A comparison with the in situ data for BY15 obtained from the Baltic NEST Institute Database (BED, Suppl. Mat. S4)
        showed a reasonable representation of the seasonal SST cycle in open ocean environments. The monthly mean climatologies
calculated by the models were well with within the standard deviations calculated for each month from the BED. Besides
        this, in the GETM_2nm the summer is colder and the winter is warmer, such that the seasonal cycle was less prominent than
        according to the BED data. The two MOM versions winter months are systematically colder but there was good agreement
        with summer data from the BED. However, the GETM_1nm best reproduced the BED cycles.

At station F9, located in the Bothnian Sea, salinities according to the SHARK data were ~3–4 g kg$^{-1}$. This range was best
        reproduced by the two GETM versions while lower salinities were obtained in the other models. Water temperatures below
        0°C were recorded in the SHARK data and in the two MOM versions, up to March/April. The weakest thermocline (i.e.,
        lowest temperature gradients) was again that of the GETM_2nm during summer.

**3.4 Long term variability in temperature and salinity**

Long-term variability was briefly assessed by the modeled time series of temperature and salinity at the same stations used to
        examine the mean climatological cycles (Figure 5). Generally, the models differed more when the inter-annual variability
        was large, exemplified by the winter SSTs at Anholt station and the summer SSTs at station F9 (Fig. 6). At stations BY2,
        BY5, and BY15, representing open sea conditions with successively larger water depths, good agreement between the
        models was obtained for both winter and summer SSTs. The inter-model standard deviation for SST increased from the
shallower site BY2 toward deeper sites BY5 and BY15, indicative of greater meteorological control at shallower than at
        deeper sites. In agreement with the analysis of the mean climatology (Fig. 4), the long-term averaged inter-model standard
        deviation of the SSTs was systematically higher during summer than in winter (Fig. 6).

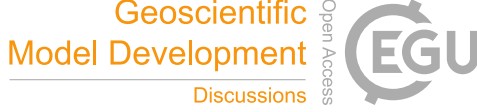

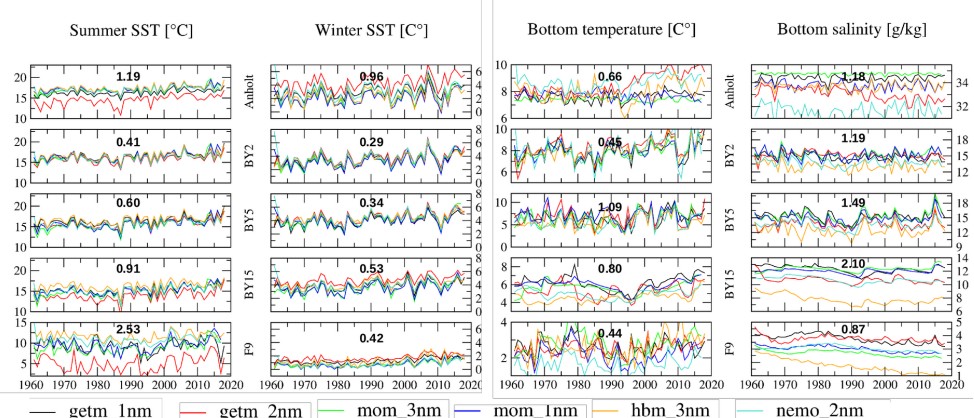

*Figure 6: Inter-model comparison of long term time series of the mean summer (JJA) and winter (DJF) SST (left panels) as well as the annual mean bottom temperature and salinity at selected sites in the Baltic Sea (right panels). Numbers in the respective panels denote inter-models standard deviations averaged over the entire period 1961-2018.*

Inter-annual SSTs co-varied quite well across the models, at all sites and for both seasons (Fig. 6). For the stations Anholt, BY15, and F9, summer SSTs were systematically lower in GETM_2nm than in the other models. Covariation was generally worse for bottom than for surface temperatures (Fig. 6). This was most obvious at the deep stations BY15 and F9, which are less well constrained by meteorological forcing. The spread in bottom temperatures at BY15 was extraordinarily low after the MBI that took place in 1993. The strength of the latter event was well reflected in all models by a corresponding shift to

higher bottom salinities, although the corresponding inter-model spread in salinities was quite large.

All in all, model agreement in bottom temperature and salinity was lowest at the deepest stations (BY5 and BY15), as indicated by the long-term averaged inter-model standard deviations. Note that nearly no inter-annual variability in bottom salinity at Anholt was recognized by MOM_3nm and the mean salinity was higher than in all other models (Fig. 6). This

suggested a more or less stable inflow of saltwater from the North Sea into the Kattegat.

The dynamics of MBIs as reflected in the deep salinity at BY15, accounted for an inter-model spread that was by far the largest (Fig. 6). While in the simulations, at least those for the decades after 1990, individual inflows were consistently recorded (although with varying amplitude), the first ~30 years may have been influenced by long-term model drifts. This

was especially the case for models in which the mean equilibrium state strongly differed from the initialization state, as occurred in the HBM_3nm. Comparison with the high-resolution in situ data from the BED showed that the results of the two MOM versions and GETM_1nm were closest to the observations (Suppl. Mat. S5). However, the two MOM versions apparently underestimated low-amplitude variations, as indicated by its relatively smooth curves, particularly during the early decades.


Station F9 is located farthermost from the overflow region and its inter-annual variability is accordingly low. A notable drift over the entire period was determined in the HBM and may have been related to differences in runoff forcing or to physically and numerically induced mixing that was too large in that run (Burchard and Rennau, 2008).

### 3.5 Brief assessment of model spread of extreme temperatures

The oceanic and atmospheric models applied in climate sciences are typically developed to reasonably reproduce long-term temperature climatologies averaged over several decades, whereas extreme temperatures are less often considered. However, the BMIP will also investigate the impact of climatic extremes and short-term events, such as heat waves (e.g. Suursaar, 2020).  The inter-model standard deviation for the 5th, 25th, 50th, 75th, 85th, 95th, and 99th percentiles of temperature averaged over the whole Baltic Sea are presented in Figure 7a which clearly shows that the standard deviation, and thus model
uncertainty, increases at high temperature regimes. Again, this conclusion could be drawn because atmospheric forcing was the same in all models, thus further demonstrating the added value of the BMIP.

Model spread with respect to water depth is shown in Figure 7b. A more or less linear increase with depth can be seen that is largest in the bottom layer. This was not unexpected, as deep-water properties are less constrained by atmospheric forcing such that initialization, model numerics, and the parameterization of sub-grid processes become more important


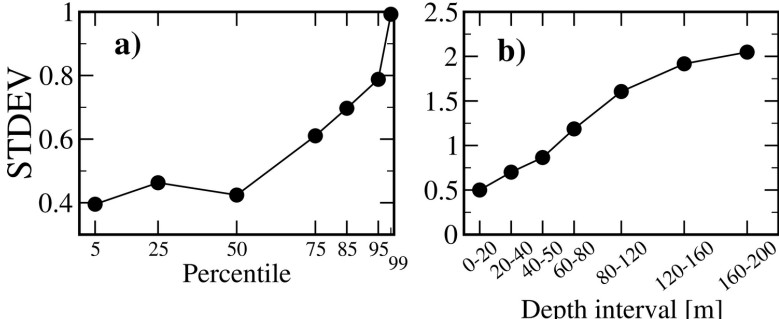

Figure 7: a) Inter-model standard deviation calculated from the 5th, 25th, 50th, 75th, 85th, 95th, and 99th percentiles surface temperatures. The percentiles represent area averages over the whole Baltic Sea. b) Inter-model standard deviation of depth-interval-averaged water temperature. Standard deviations are calculated from spatial averages over the whole Baltic Sea from each of the six models. The analysis covered the period is 1990 – 2007.

## 4. Topical case studies

### 4.1 Marine heat waves

Climate warming increases the risk of extreme events in ocean climate. For example, MHWs in the world ocean are
expected to be more frequent and intense in a warmer climate (Oliver et al., 2019). Due to its low water volume and limited exchange with the open ocean, the Baltic Sea is especially sensitive to external changes in the heat supply. Unlike the North Sea, which fully mixes during winter and is well ventilated by waters from the North Atlantic within a few years, in the Baltic Sea the perennial halocline limits heat exchange between the surface and deeper layers. Accordingly, larger and smaller warming of the surface and sub-halocline layers, respectively, can be expected. In the Baltic Sea models analyzed
herein, this was well reflected by the larger increase in surface than in bottom temperatures since the mid 1980s(Fig. 8). Moreover, extreme SSTs can increase more than mean SSTs. As shown in Table 2, higher warming trends for the annual maximum temperature than for the annual mean temperature were determined by all of the models. Likewise, the higher cross-model standard deviation in the maximum temperature trends than in the mean temperature (Table 2) implied higher





uncertainties in the high-temperature regime. These results highlight the need for studies on the processes leading to extreme

SSTs in the Baltic Sea.

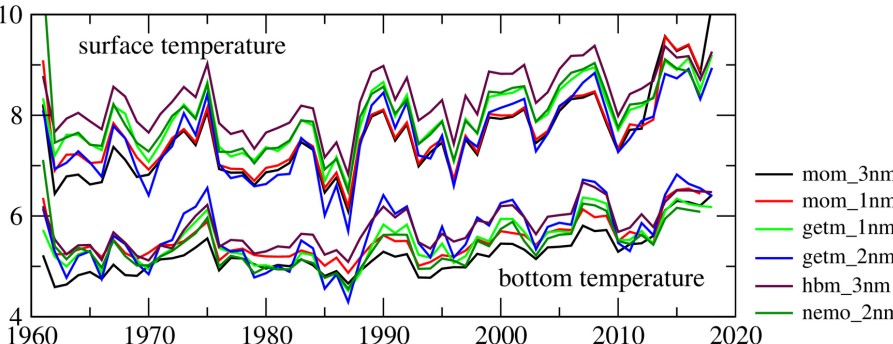

*Figure 8: Annual mean surface and bottom water temperatures averaged over the entire Baltic Sea.*

| Model | Yearly mean trend [K/yr] | Yearly maximum trend [K/yr] | Difference max minus mean trend [%] |
|---|---|---|---|
| HBM_3nm | 0.026 | 0.038 | 46.15 |
| GETM_2nm | 0.034 | 0.052 | 52.94 |
| GETM_1nm | 0.029 | 0.044 | 51.72 |
| MOM_1nm | 0.034 | 0.048 | 41.18 |
| MOM_3nm | 0.034 | 0.059 | 73.53 |
| NEMO_2nm | 0.027 | 0.036 | 33.33 |
| STD | $3.8*10^{-3}$ | $8.7*10^{-3}$ | |

*Table 2: Comparison of yearly mean and maximum temperature trends averaged over the whole Baltic Sea*

Figure 9 shows the yearly mean area affected by different classes of MHWs. The models were compared with the reanalysis data set covering the reference period 1970–1999 (Liu et al., 2017), which was characterized by two distinct maxima, in 1975 and 1990, when areas > 125 000 km$^2$ were affected vs. < ~25 000 km$^2$ during the intervening period. These two peaks

were well reproduced by the models. The longer record of the BMIP models allowed the identification of pronounced periods of high MHW extensions, with peaks occurring in 1975, 1990, 2002, 2009, 2016, and 2018 thus pointing to roughly decadal variations until 2002 and the potential increases due to climate warming afterwards. The weak imprint of MHWs in the second half of the 1970s and 1980s might be related to the extraordinarily low North Atlantic SSTs recorded during those years (Kushnir et al. 1994). In all of the models there was a trend toward more extended MHWs after ~1990, consistent with

the climate warming trend over that same time (e.g., Dieterich et al., 2019; Gröger et al. 2019; Meier et al., 2022; Placke et al., 2021; Dutheil et al, 2021). Before ~2000, MHWs were rarely above the moderate class whereas strong MHWs (class II) became more prominent thereafter. The inter-model differences were rather low, most obviously during the early period, but the MOM simulations yielded very highly extended MHWs especially during the past decade.

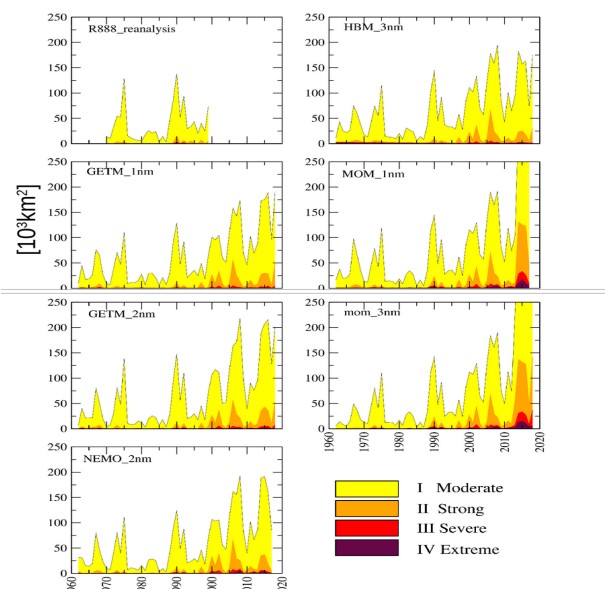

*Figure 9: Yearly average spatial sea surface extent of MHWs over the entire Baltic Sea. The reanalysis data set refers to Liu et al. (2017). Classification was done after Hobday et al. (2018).*

Next, MHW frequency was analyzed, by counting the number of periods with at least five consecutive MHW days (class I or higher). MHWs separated by only one or two days were counted as one MHW. Determinations were done separately for the 25-year periods 1965–1989 (early period) and 1994–2018 (late period). The results are shown in Figure 10. For the early period (Fig. 10a), all models indicated that MHWs were most frequent in the Kattegat, the Arkona Basin, and in Bothnian Bay, i.e., shallow areas or areas with seasonal sea ice cover. The largest inter-model differences as indicated by the ensemble

standard deviation (Fig. 10a) occurred in the Bothnian Bay and in the easternmost Gulf of Finland and were likely related to differences in the modeled sea ice cover, which affected ocean-atmosphere heat exchange.

The average MHW duration varied spatially between 8 and 25 days, as shown in Figure 10b. Longest MHWs occurred in the Bothnian Bay. The ensemble spread was highest in the Bothnian Bay, and locally elevated in the Bothnian Sea, and the

central Baltic proper, as indicated by the ensemble standard deviation. In shallow regions and along the coasts, MHW duration was consistently short, as these areas are more prone to variable meteorological forcings that may disrupt MHWs, such as storm events or cold-water intrusions from the open sea. As MHWs of longer duration will ultimately limit the number of possible MHWs within a given time period, the models showed a negative relationship between average MHW length (Fig. 10b) and MHW frequency (Fig. 10a). However, the correlation between the duration and number of MHWs

differed considerably between models, with $r=-0.65$ for MOM_3nm, $r=-0.57$ for MOM_1nm, $r=-0.47$ for GETM_2nm, $r=-0.35$ for NEMO_2nm, $r=-0.28$ for GETM_1nm, and $r=-0.02$ for HBM_3nm (averaged in each case over the Baltic Sea).

In the late period 1994–2018, MHWs were almost uniformly more frequent and of longer duration (Fig. 10c,d). Common to all models is the strong increase in the Gotland Basin where relative increases exceeded 200%. Both MOM versions showed



an extraordinary increase in average MHW duration, thus offering an explanation for the extraordinarily large spatial extension of MHWs that occurred during the last decade (Fig. 9), as a longer duration favors a larger spatial extension and vice versa. In the HBM_3nm and GETM_1nm, the changes in the frequency and duration were smaller than in the other models.

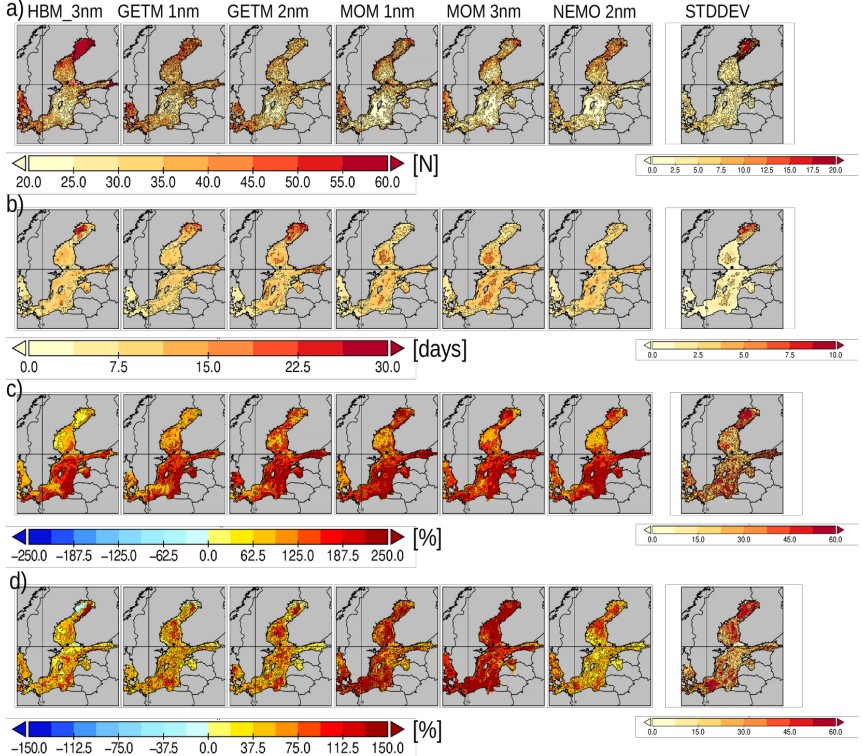

*Figure 10: a) Total number of MHWs with a duration of at least five consecutive days (class I or stronger) during the period 1965–1989. b) Average MHW duration (class I or stronger). c) Relative change in the number of MHWs between the period 1994–2018 and the period 1965–1989. d) same as c) but for the average MHW duration.*

**4.2 Coastal upwelling**




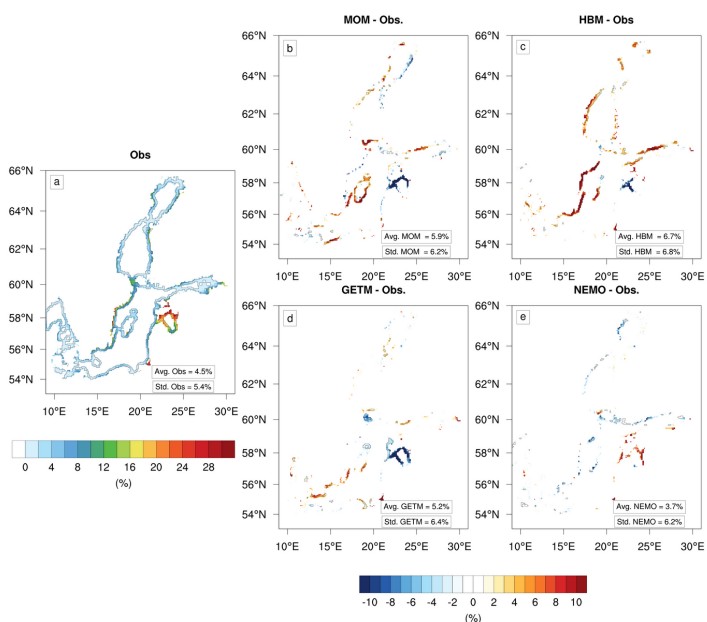

*Figure 11: Annual upwelling frequencies (in %) in (a) the observations and (b–e) the errors made by models (b) MOM_3nm, (c) HBM_3nm, (d) GETM_1nm and (e) NEMO_2nm. The average and standard deviation are shown in the bottom-right corners.*

Figure 11 displays the annual upwelling frequencies according to the BSH satellite data and the deviation therefrom in each simulation. In the former, the average annual upwelling frequency over the Baltic Sea was 4.5%. Upwelling areas were

concentrated along the Swedish coast and in the Gulf of Riga, where upwelling frequencies can exceed 20%. Along the others coastal regions, the annual upwelling frequency was < 10, resulting in a spatial standard deviation of 5.4%.
All hindcast simulations except that of NEMO_2nm overestimated the annual upwelling frequencies compared to the observations, but with some discrepancies. Thus, according to HBM_3nm, MOM_3nm, and GETM_1nm the annual upwelling frequency was 6.7%, 5.9%, and 5.2%, respectively. In the NEMO_2nm simulations, the annual upwelling

frequency was 3.7% and thus underestimated. Overall, the spatial pattern of the annual upwelling frequencies was well represented in all hindcasts, with a similar bias pattern between the models except NEMO. Hence, GETM_1nm, MOM and HBM_3nm tended to underestimate the upwelling frequencies in the Gulf of Riga and to overestimate them along the Swedish coast, around Gotland, and in the Gulf of Finland (Fig. 11). The opposite spatial bias pattern was determined for NEMO_2nm. Thus, overall, the spatial standard deviation was overestimated by 6.2% (MOM_3nm and NEMO_2nm) to

6.8% (HBM_3nm) compared to 5.4% in the observations.

Figure 12 shows the weekly and inter-annual variations in the observed and modeled upwelling frequencies. According to the observations, the weekly upwelling frequency averaged over the Baltic Sea varied from 2 to 7%, with minimum values occurring at the end of winter and in early spring, i.e., between weeks 5 and 15, and maximum values at the end of the year,

around week 50. From spring to fall, the upwelling frequency was also high (reaching 6%). From 1993 to 2010, the annual upwelling frequency was characterized by a strong inter-annual variability, varying by a factor of 2 (from 3% to 6%), with a





frequency between 3 and 6 years (determined by wavelet analysis).

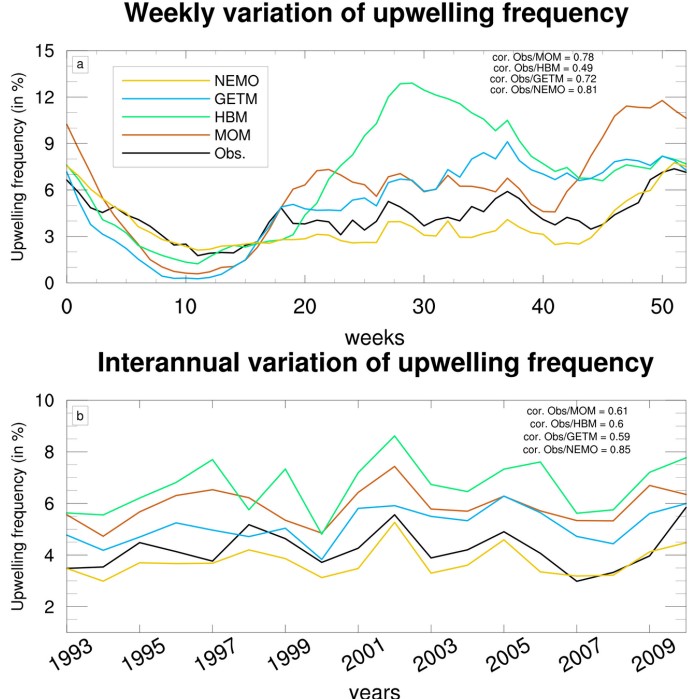

*Figure 12: Weekly (a) and interannual (b) variations in upwelling frequencies (in %) according to observations (black lines) and to the models MOM_3nm, HBM_3nm, GETM_2nm and NEMO_2nm (red, green, blue and yellow lines respectively). The correlations between observations and models are shown in the top-right corners.*


The mean weekly variations in upwelling frequency were well modeled, with the correlations between observations and the models ranging from 0.49 (HBM_3nm) to 0.81 (NEMO_2nm) although the models tended to overestimate the amplitudes. The GETM_2nm, MOM_3nm, and HBM models underestimated the upwelling frequencies between weeks 5 and 15 and overestimated them from week 20 to the end of the year. These biases were reduced in NEMO_2nm, in which the weekly

variation was similar to that in the observations. The mean weekly amplitude was thus 11% in MOM and HBM_3nm, 9% in GETM_2nm, and 5.6% in NEMO_2nm, compared to 5.6% in the observations. The inter-annual variability was also well modeled, with correlations between observations and the models of ~0.6 for GETM_2nm, MOM_3nm and HBM_3nm and 0.85 for NEMO_2nm. The overestimation or underestimation of annual upwelling frequencies shown in Figure 11 were consistent with the results presented in Figure 12. In contrast to the biases in the mean weekly upwelling frequencies, those

in the inter-annual upwelling frequencies were almost stationary.

To conclude, the biases in the hindcast simulations of GETM_2nm, MOM_3nm, and HBM_3nm were similar and characterized by an overestimation of the annual mean upwelling frequency and of the spatial variability while opposite and



smaller biases were obtained with the NEMO_2nm simulation. The upwelling frequency was overestimated around Gotland,
along the southern Swedish coast, and in the Gulf of Finland and underestimated in the Gulf of Riga. The weekly amplitude
of the upwelling frequency was also overestimated but the inter-annual variability was well simulated. Overall, opposite
conclusions were derived from the NEMO_2nm simulation.

The upwelling analysis highlighted the differences in the BMIP models and thus the importance of systematic inter-model
comparisons. Deeper analyses, for instance, inter-model comparisons aimed at determining the contributions of the
individual mechanisms responsible for upwelling events (e.g., Ekman pumping, hydrodynamic circulation), could provide
insights into the reasons for the differences in upwelling (e.g., parameterization of internal waves or air-sea fluxes).

### 4.3 Water column stratification

The Baltic Sea is stratified by both a permanent halocline and a seasonal thermocline. The depth and strength of these
"clines" vary spatially and over time, with the summer thermocline developing in late spring and lasting until the transition
summer/autumn. A good model representation of thermocline development is therefore especially important for
biogeochemical models, as it must accurately depict the timing and intensity of the spring bloom.

As an example, the typical climatological development of the summer thermocline as determined in the BMIP models is
shown in Figure 13.  For comparison, the depths included in the figure are the same as those calculated from the observed
vertical profiles using the ICES database. Point observations were grouped with vertical profiles based on a common
latitude, longitude, and date. Vertical profiles less than 60 m deep and that included jumps in the vertical coordinate > 5 m
were excluded from the analysis. For the period 1960–2018, the cline depths for each of the observed profiles were
calculated. Finally, a monthly climatology was created by taking the median of all values in a calendar month, for every box
of a 1- × 1-degree horizontal grid. The median rather than the mean was used to reduce the sensitivity to outliers. Finally, at
least five values per box were required to consider the median as valid.

The results showed that, in April, all of the considered models overestimated the thermocline depth, which in the southern
part of the central Baltic Sea was ~30 m. The overestimation was lowest in the coarser models (HBM_3nm and the
MOM_3nm) and in NEMO_2nm. Near-coastal regions, where low values of the thermocline depth were determined by the
models, were excluded from the observational climatology because of the required minimum depth of 60 m. The
observations showed that the summer thermocline formed already in May, with the thermocline depth dropping to ~20 m.
This was reasonably captured by the HBM_3nm and NEMO_2m models, whereas in the GETM_1nm and especially the two
MOM models thermocline shallowing was delayed. In June, the models determined a reduction of the thermocline depth to
~20 m, which was a slight underestimation compared to the observed value. The GETM_1nm model differed from the others
as it showed a clearly enhanced thermocline depth in the deeper parts of the southern Baltic proper.
Further detailed analyses of model output may reveal the reasons underlying the difference in the timing of thermocline
formation despite identical atmospheric forcing.


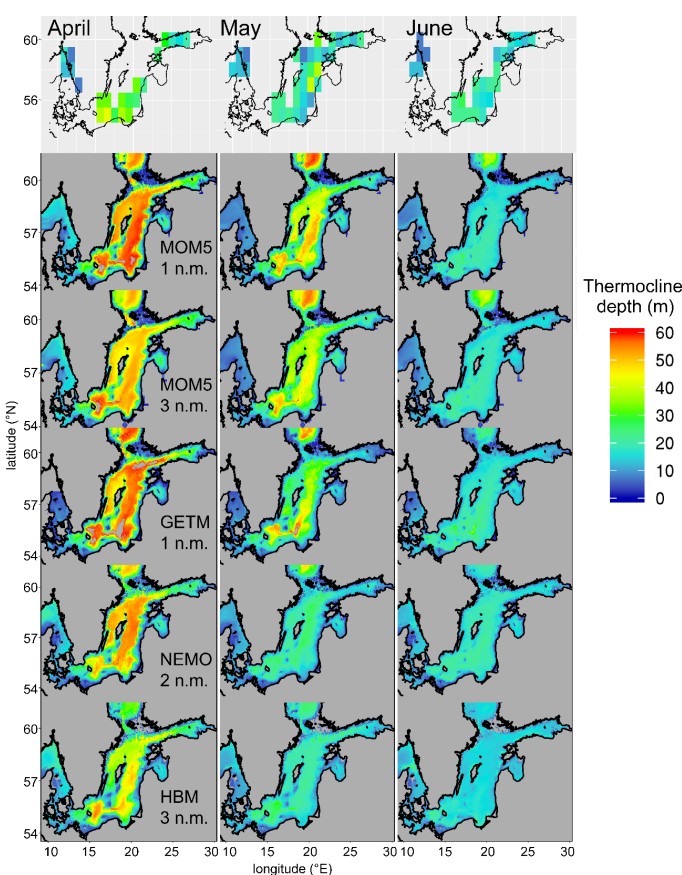

*Figure 13: Thermocline depth as derived from ICES observational data (uppermost row) and from different BMIP models. Grey areas in the ICES maps indicate a lack of data, in the BMIP model maps they denote values above 60 m. Note, the maps are cut off in the north due to lack of sufficient ICES data for the cline calculation.*

## 5. High resolution modeling and the BMIP

In addition to the inter-model comparisons presented herein, a very high-resolution model for the central Baltic Sea is under development within the framework of the BMIP collaboration. The aim of this computationally challenging project is to investigate large-scale ocean circulation with respect to the role of mesoscale and sub-mesoscale processes. Scientific questions regarding the importance of eddies and other small-scale processes in the exchange of dissolved nutrients and toxins between the coastal zone and the open sea will be examined as well



The model setup is built on the GETM source code and the model domain covers most of the Baltic Sea, including the
Kattegat and Danish straits and both the Gulf of Finland and the Gulf of Riga. The northernmost part of the Baltic Sea, i.e.,
the Gulf of Bothnia, consisting of the Bothnian Sea and Bothnian Bay, has been replaced by an open boundary

The importance of high-resolution simulations is illustrated in Figure 14, in which different parameters simulated with low
(1 nm) and high (250 m) resolution are compared. In general, large-scale patterns were well simulated by both. In each case
there was a strong south-to-north gradient in the simulated surface temperature fields, with the highest (lowest) temperatures
in the southern (northern) part of the Baltic Sea. In addition, in the north-western Baltic proper, a large patch of cold water of
upwelling origin along the Swedish coast and advected from the Gulf of Bothnia was well visible in both simulations. In
contrast to the situation in the north, warm water in the coastal regions of the southern and eastern Baltic proper were
determined in the two simulations. The largest differences between the results obtained at high vs. low resolution involved
several details of the simulations. First, the low-resolution model was generally unable to produce strong lateral gradients in
open sea areas, except in cases in which strong fronts already occurred due to mesoscale activity (upwelling along the
Swedish coast). Second, eddy activity in the open sea was much smaller in the 1-nm simulations than in the 250-m resolution
simulations, with much weaker (geostrophically balanced) eddies in the low-resolution run, whereas in the high-resolution
run much stronger and ageostrophic eddies (Rossby number > 1) were produced, both in the coastal area and in the open sea.
Weaker eddy activity in open sea areas with low resolution were also visible in the spatial maps of kinetic energy.

The overall purpose of the high-resolution simulations was to analyze the role of eddies in the Baltic Sea (e.g., Lips et al.,
2016; Väli et al., 2017; Väli et al., 2018). Several studies of the mean circulation (e.g., Lehmann et al. 2002; Meier, 2007;
Placke et al., 2018) and of long-term nutrient transport (e.g., Eilola et al., 2012) performed using low resolution models are
available and they provide evidence of large-scale gyre structures with strong persistent currents in the eastern Gotland Basin
and of an overall estuarine circulation in the Baltic Sea. However, these models were largely eddy-permitting rather than
eddy resolving. Vortmeyer-Kley et al. (2019a,b) attempted to quantify the number of eddies and their lifetime using higher-
resolution models while Zhurbas et al. (2018) provided a qualitative comparison of observed and simulated eddies. The
importance of eddies in transport within the Baltic Sea is therefore still unclear. Long-term, high-resolution simulations that
allow the representation of sub-mesoscale structures are likely to yield important information.



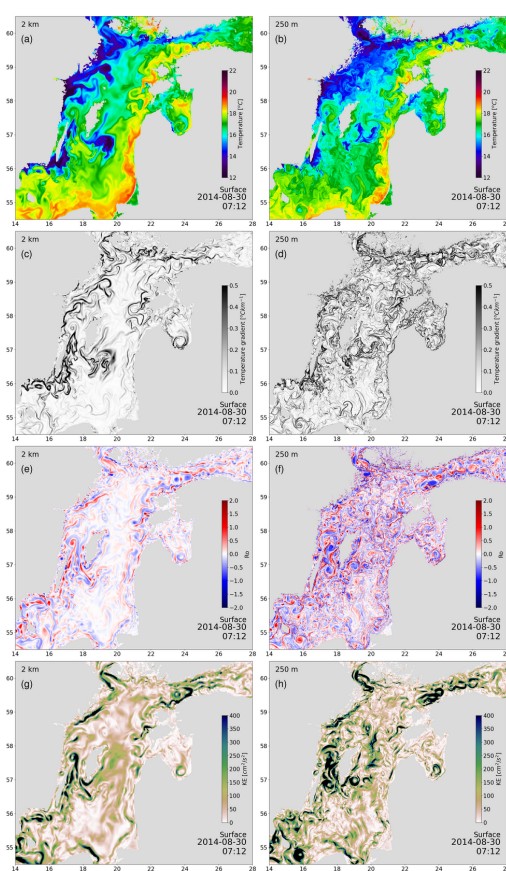

*Figure 14: (a,b) Snapshot of surface layer temperature [°C], (c,d) temperature gradient [°C km⁻¹], (e,f) Rossby number, and (g,h) kinetic energy [cm⁻²s⁻²] as obtained from the low-resolution (1nm, left) and high-resolution (250 m, right) simulation for the Baltic Sea.*

## 6. Summary and Conclusions

The BMIP provides almost 60 years of physically consistent data on meteorological and hydrological forcing, for use in
Baltic Sea ocean modeling. This study, the first systematic model inter-comparison, revealed marked local to regional model
differences in simulated temperature and salinity, in vertical thermal and haline stratification, and in distinct climate and
environmental indices (e.g., heat waves, upwelling, stratification). Our results thus emphasize the role of internal model
dynamics, in addition to external forcing, and thereby highlight the benefit of coordinated model comparisons, such as those
within the BMIP, to disentangle causes of model differences.

The spread in the six different models, and thus the uncertainty related to internal model dynamics, was larger in the extreme
high-temperature regime than in the average temperature regime (i.e., for higher percentile temperatures). In all models,







linear warming trends were higher for annual maximum than for annual mean SSTs, but the uncertainty in annual maximum temperature trends was twice as high. Likewise, the models differed more with respect to simulated bottom water temperatures than to SSTs. This was expected, as bottom waters are less constrained by meteorological forcing such that internal model dynamics are more important. However, for sub-halocline waters, longer-term drifts can be expected when the model's internal equilibrium state strongly differs from its initialization state. This is especially the case for operational models, which are not designed to run in the free climate mode without massive data assimilation. Furthermore, particularly high uncertainties were found in the northern Baltic Sea, in line with previous studies (Eilola et al., 2011; Placke et al., 2018). This was very likely related to the different employed sea ice modules and thus to the differences in air-sea heat fluxes.

Generally, the inter-model spread in SST was larger in summer than in winter (Fig. 4). During summer, the presence of a strong thermocline reduces the effective heat capacity, resulting in a larger correlation between the meteorological forcing and the SST. Consequently, slight differences in the depth and intensity of the thermocline can greatly affect the thermal state of the water column, which translates as a large model spread. However, in shallow regions along the coast, where a stable thermocline cannot develop, a rapid adaption to the (same) meteorological boundary takes place and strongly diminishes inter-model spread. By contrast, strong oceanic heat loss together with strong wind and convective mixing during winter increases the effective ocean heat capacity, dampens temperature variations, and minimizes inter-model spread. The large inter-model spread in summer SST in the northern Baltic Sea can probably also be explained by the different melting rates and sea ice break-up dates.

Analysis of the long-term variability revealed better agreement between models for areas where the variability is low, such as in the Arkona Basin or Bornholm Basin, than for areas with high interannual variability, e.g., the Bothnian Sea (Fig. 6). Models that were primarily developed for operational services typically run only for short periods (i.e., days to a couple of months) and thus have not been validated in long-term simulations for multiple decades. Consequently, these models often show significant drifts in long-term runs and suffer from considerable biases regarding near-bottom salinity (e.g., Hordoir et al., 2019). In this context, the BMIP seeks to promote knowledge exchange across different model platforms.

We also investigated selected topical case studies, such as MHWs, coastal upwelling, and stratification, in some of the models. The aim of these analyses was to illustrate the impact of model biases on, for instance, simulated extremes and to highlight still-open questions hindering an understanding of all of the models' shortcomings. For example, in all of the models the thermocline was substantially deeper than that calculated from observational data for early spring (April and May). However, the bias was reduced when the thermocline intensified during June. In GETM_1nm, MOM_1nm, and MOM_3nm, the formation of the thermocline was delayed compared to the other models and to the observations.

Analysis of MHWs revealed substantial inter-model differences in their extension, frequency, and duration. Nonetheless, all of the models showed more frequent as well as longer and spatially more extended MHWs during the past three decades. Generally, MHWs were more frequent near the coasts and in shallow areas (Kattegat, Danish Straits), as both are more prone to variable meteorological forcing. However, regional differences among the models were identified, especially in regions seasonally covered by sea ice (Bothnian Sea, Bothnian Bay).





Upwelling frequencies were mostly overestimated in the models (GETM_2nm, MOM_3nm and HBM_3nm), in particular along the Swedish coast, around Gotland, and in the Gulf of Finland. Lower upwelling frequencies were registered in the

Gulf of Riga. Compared to the other models, in NEMO_2nm the biases were reduced and of opposite sign.

To investigate the effect of the grid resolution on model performance, a first set of ultra-high resolution simulations resolving sub-mesoscale features was carried out within the BMIP, using the GETM model platform and comparing snapshots of simulations with horizontal grid resolutions of 1 nm and 250 m. Generally, lateral SST gradients were much

stronger in the 250-m version in the open sea. This was accompanied by higher eddy activity, which is less constrained by geostrophy. The difference was less pronounced in coastal regions affected by upwelling, such as the Swedish coast. As sub-mesoscale fronts are connected with large vertical velocities, an impact of the high-resolution simulation on the mixing of water masses can be expected. Furthermore, the simulation of strait-flow dynamics and overflows of gravitationally driven dense bottom currents might be improved by a better representation of physical processes and bottom topography. However,

our simulations were too short to investigate these effects systematically, thus highlighting the need for further investigations.

**Code and Data availability**

"All data forcing/boundary data necessary to carry out a BMIP hindcast simulation along with detailed instructions and code

can be downloaded from the BMIP web portal at: https://baltic.earth/working_groups/model_intercomparison/index.php.en. The atmospheric and hydrological forcing data can also be downloaded from the Copernicus Climate Change Service information [2019], see https://cds.climate.copernicus.eu/cdsapp#!/dataset/reanalysis-uerra-europe-complete?tab=overview, and from http://doi.io-warnemuende.de/10.12754/data-2022-0005, see also the detailed report on the river discharge data at http://doi.io-warnemuende.de/10.12754/msr-2019-0113, respectively. The model codes of the four Baltic Sea models, i.e.

MOM, NEMO and HBM, are available at https://zenodo.org/record/6560174#.YsKpiYTP1PY, https://doi.org/10.5281/zenodo.1493116, https://doi.org/10.5281/zenodo.6769238, respectively. The GETM code is available as supplementary material S5 and the BMIP instructions are available a supplementary material S6.

The data sets generated during and/or analyzed during the current study are available from the corresponding author upon

reasonable request. Numerical model codes are available from the respective literature and corresponding first author.

**Author contributions**

MG led the study, performed most of the analysis, and wrote most of the text. MM launched the BMIP project and led the design of the BMIP protocoll. CD analyzed upwelling and wrote the respective section. HR analyzed stratification and wrote

the respective section. GV and MM anlyzed and wrote the section on high resolution modeling. Individual model experiments were carried out by UG, TN, FB, S-EB, MH, and JS. All authors contributed to vigorous discussions about the interpretation of results and data analysis.

**Competing interests**

The first author declares that none of the authors have any competing interests.

**Acknowledgments**

The research presented in this study is part of the Baltic Earth program (Earth System Science for the Baltic Sea Region, see http://www.baltic.earth). We thank the Federal Maritime and Hydrographic Agency Hamburg and Rostock (BSH) for



financing and for supporting the operation of the MARNET stations in the western Baltic Sea. Temperature and salinity data used in the evaluation are open access and were extracted from the Baltic Environmental Database (BED, http://nest.su.se/bed) at Stockholm University; all data-providing institutes (listed at http://nest.su.se/bed/ACKNOWLE.shtml) are kindly acknowledged. GETM and MOM model development and simulations were performed with resources provided by the North-German Supercomputing Alliance (HLRN). Model simulations with

NEMO, provided by the Swedish Meteorological and Hydrological Institute, Sweden, (Sveriges Meteorologiska och, Sveriges Meteorologiska och Hydrologiska Institut), were conducted on the Linux cluster Bi operated by the National Supercomputer Centre (NSC), Sweden (http://www.nsc.liu.se/). Germo Väli was supported by the Estonian Research Council (grants no. IUT19-6 and PRG602) and by the Leibniz Institute of Baltic Sea Research during his stay at the IOW in Warnemünde. Computational resources from HLRN and Tallinn University of Technology are gratefully acknowledged. We

thank Sergei Zhuravlev for providing the daily Neva runoff data and Uwe Schulzweida, the R Core Team, and the Unidata development team (and all involved developers/contributors) for maintaining the open source software packages Climate Data Operators (cdo), the statistical computing language R, and netCDF, respectively. The E-OBS dataset and the data providers in the ECA&D project (https://www.ecad.eu) are acknowledged.

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
