# Peer review of "The Baltic Sea model inter-comparison project BMIP – a platform for model development, evaluation, and uncertainty assessment"

_Geoscientific Model Development, 2022_

## Referee Comment (RC1)

**General comments**

The paper addresses relevant scientific modeling issues using a never-before-used and undeniably rigorous protocol for comparing numerical models of the Baltic Sea dynamics. This method is useful in determining the discrepancies between the models because the same forcings were applied to each of the models, thus allowing the discrepancies between the models to be interpreted as being due to the model's own configuration or grid resolution. The authors have well introduced the lack of inter-comparison studies of regional models against other (global) models and the need to do so for the Baltic Sea and the North Sea. To justify the importance of this exercise, they highlighted the complexity of the site and the diversity of dynamic models used to simulate, among others, the general circulation. A state of the art of the models used in the study area is quite complete, it is presented at the beginning of the article. The detailed method is available on the project's website and allows for identical replication of the experiments. The added value is in the potential reproducibility of the method to other marginal seas. Thus, the technical approach is clearly explained with some exceptions that will be mentioned in the "Specific Comment" section. The overall structure of the paper and its presentation make it clear and easy to read. However, some parts of the results need revision which I detail in the "Specific Comment". In addition, many of the figures need to be reworked. However, the main messages of the publication are clear but the results lack discussion and consideration of related work.

**Specific comment**

For these comments I list them in the same order as in the publication.

**Abstract**
It is detailed and includes the main results. The absence of information about salinity is deplored.

**Methods:**

This part is very clear, which makes it easy to notice the few missing information.

**Runoff** used for this study are clearly referenced but the way they are implemented is not explained, if the runoff is added to the first mesh from the coast or diffused over several meshes, if they are applied only on the surface or on the whole water column?
Moreover, contrary to the choice of atmospheric forcing which is justified, the choice of runoff is not explained.

**Spin-up** : There are many references to model stability in the article, however, in the supplementary material there is no figure showing the stability of each of the models. Although the recommendations are clear, they are not explained. Why is it recommended to run the simulations again in July 2004 and not another month? What initialization was used to start the spin up runs?

Also implied by these comments is the issue of applying the same spin-up for all models despite differences in grid resolution and turbulence schemes.

**Analysis methods :** It may be of interest to indicate the error associated with the post-processing of AVHRR data.

Specifically, the upwellings detection method used in this study is that of Lehmann et al, 2012 despite the biai from the position of the coastlines whose axis is different from the East/West axis. Why this choice of method? Why not use another method as described in Schlegetk & Smit; 2018 and Abraham, Schlegetk & Smit; 2021.

**Results:**

In the introduction of the results, it is stated that different runoffs were for the HBM model. This part should be in the material and method explaining the reason for this choice and specifying which runoff were used.

**In the first part of the results** the role of thermoclyne formation in the sensitivity of the SST to variations in meteorological forcing is stated but sparsely discussed. This lacks discussion and bibliographic references.

**The section dealing with seasonality** needs to be restructured. Suggestion: Discuss the divergences of the models, station by station, with respect to temperature and then do the same for salinity, in the same way as the introduction to Figure 5.

Indeed, the paragraphs introducing the stations describe sometimes the variability of temperature, sometimes that of salinity.

The discussion of temperature variability for the Nemo model is missing.

**Long term variability:** In this part we still refer to the stability of the models. It is therefore necessary to put the figures that illustrate these remarks in the publication.

Also, in this and several times, it is referred to divergences of models because of their different management of ice modules, what about turbidity that can limit the heat flux?

**Marine heat waves:** Figure 8 with Table 1 again confirm what was explained in section 3.5 without adding additional information. It would be interesting to compare the models with the data in Figure 8 to see which model is closer to the observed extreme values and not just that the models diverge more for extreme temperatures.

**Upwelling:** In figure 11, the GMT_1nm model is analyzed, while in figure 12, GMT_2nm is analyzed. Why this choice and why not treat the outputs of the MOM_1nm model in the upwellings analysis?

**Water column stratification:** This section ends with "Further detailed analyses of model output may reveal the reasons underlying the difference in the timing of thermocline formation despite identical atmospheric forcing." What do you suggest? This section should be discussed with references.

**Summary :**

In the conclusion, taking Hordoir et al., 2019 as an example of non-validated models in long-term simulations is not accurate because, in the first instance, the HBM model was chosen in the experiments as an example of an operational model. Furthermore, in Hordoir et al. 2019, the model is described as one that allows research on long-term simulations as much as on operational applications and whose simulations are devoid of data assimilation.

Finally, salinity has once again been little discussed even though it is strongly impacted by runoffs, MBI…

**Technical corrections**

**Reference error**
This is not an exhaustive list

Name written differently:

Meier HEM, Döscher R, Coward AC, Nycander J, Döös K: RCO—Rossby Centre regional Ocean climate model: model description (version 1.0) and first results from the hindcast period 1992/93. Reports Oceanography No. 26, SMHI, Norrköping, Sweden, p 102, 1999.

Meier, H. E. M., and S. Saraiva : Projected Oceanographical Changes in the Baltic Sea until 2100. Oxford Research Encyclopedia of Climate Science, online publication date:. DOI: 10.1093/acrefore/9780190228620.013.69, 2020.

Meier, H.E.M., Dieterich, C., Gröger, M.: Natural variability is a large source of uncertainty in future projections of hypoxia in the Baltic Sea. Commun Earth Environ 2, 50 (2021). https://www.nature.com/articles/s43247-021-00115-9, 2021a.

Listed as duplicates:

Meier, H. M., Höglund, A., Döscher, R., Andersson, H., Löptien, U., & Kjellström, E. (2011). Quality assessment of atmospheric surface fields over the Baltic Sea from an ensemble of regional climate model simulations with respect to ocean dynamics. Oceanologia, 53, 193-227.

**Figures**
Fig.3 Use a different color palette for absolute values and differences for better readability. Figure 3.e does not seem to have a colorbar with such a layout. Correct the extends of the colorbars that look truncated.

Fig.5 : Negative temperatures referred to in the text are not displayed on the scale
      -Fig.5.a put the colorbar at the end of the figure horizontally
      - Use the same width for all colorbars
      - Center the station names
      -Fig.5.b set the colorbar below each figure concerned and horizontally

Fig.10 : Reorganize the colorbars, the choice of palettes is not appropriate, the Fig10.c and Fig.10.d seem to have the same color palette

---

## Author Comment (AC1)

**Reply to reviewer 1**

**General comments**
**The paper addresses relevant scientific modeling issues using a never-before-used and undeniably rigorous protocol for comparing numerical models of the Baltic Sea dynamics. This method is useful in determining the discrepancies between the models because the same forcings were applied to each of the models, thus allowing the discrepancies between the models to be interpreted as being due to the model's own configuration or grid resolution. The authors have well introduced the lack of inter-comparison studies of regional models against other (global) models and the need to do so for the Baltic Sea and the North Sea. To justify the importance of this exercise, they highlighted the complexity of the site and the diversity of dynamic models used to simulate, among others, the general circulation. A state of the art of the models used in the study area is quite complete, it is presented at the beginning of the article. The detailed method is available on the project's website and allows for identical replication of the experiments. The added value is in the potential reproducibility of the method to other marginal seas. Thus, the technical approach is clearly explained with some exceptions that will be mentioned in the "Specific Comment" section.**
**The overall structure of the paper and its presentation make it clear and easy to read. However, some parts of the results need revision which I detail in the "Specific Comment". In addition, many of the figures need to be reworked. However, the main messages of the publication are clear but the results lack discussion and consideration of related work.**

We gratefully thank the reviewer for his/her thorough reading and the specific comments which we will consider and which will help to improve the manuscript significantly.

**Abstract**
**It is detailed and includes the main results. The absence of information about salinity is deplored.**

That's a good idea. We will include a short statement regarding the main salinity results.

**Methods:**
**This part is very clear, which makes it easy to notice the few missing information.**

**Runoff used for this study are clearly referenced but the way they are implemented is not explained, if the runoff is added to the first mesh from the coast or diffused over several meshes, if they are applied only on the surface or on the whole water column?**

The implementation is not specified in the protocol because the models manage this differently, as it is constrained by internal model configuration such as the vertical and horizontal grid spacing and options in the model code.

We will note the implementation for every model (GETM,MOM,HBM,NEMO) in the revised manuscript:

**HBM, MOM**: runoff is added to one coastal grid cell in the top layer
**GETM**: always in the top cell
discharge < 500 m³/s - one cell discharge near the coast.
discharge < 1000 m³/s - spread evenly over two horizontal cells near the coast
discharge > 1000 m³/s - spread evenly over three horizontal cells near the coast
**NEMO**: one cell discharge near the coast. Distributed over the whole water column.

**Moreover, contrary to the choice of atmospheric forcing which is justified, the choice of runoff is not explained.**

As there are no homogeneous river discharge datasets for the entire period 1961-2018 available and because the last years were only covered by the E-HYPE model forecast product, we merged the two E-HYPE model datasets and the observational records. At least for the basin averages the BMIP dataset is homogeneous and consistent.

**Spin-up : There are many references to model stability in the article, however, in the supplementary material there is no figure showing the stability of each of the models. Although the recommendations are clear, they are not explained. Why is it recommended to run the simulations again in July 2004 and not another month? What initialization was used to start the spin up runs?**

The BMIP protocoll provides no initial data for the start of the spinup. As the Baltic Sea has an overturning time of about 30 years, BMIP gives the conservative recommendation for a 1961-2004 (44 years, thus > than the Baltic Sea overturning time) spinup to reach an equlibrium where potential drifts can be minimized. BMIP recommends to start the production runs in mid-summer as in this season Major Baltic Sea salt inflows (MBI) from the North Sea are extremely unlikely.
We will make this more clear in the revised version.

**Also implied by these comments is the issue of applying the same spin-up for all models despite differences in grid resolution and turbulence schemes.**

This is correct. We will include a comment that model internal turbulence schemes and resolution may influence the time the model reaches equilibrium. That's why  we recommend an at least  44 year spinup duration (>overturning time) which is a compromise between costs to drive the model and the minimization of potential drifts.

We will make this more clear in a revised version.

**Analysis methods : It may be of interest to indicate the error associated with the postprocessing of AVHRR data.**

Thank you for your comment. To address it, we downloaded the raw AVHRR dataset and compared it with the post-processed dataset (Fig R1). This figure shows that the raw AHRR dataset underestimates the upwelling frequency by 0.9% but overestimates the spatial variability because it overestimates the frequency in Bothnian Bay. This result is consistent with the principle of post-processing as it unmasks regions misidentified by the cloud detection algorithm. We added this paragraph in the ms to discuss this point: "A comparison between raw AVHRR dataset and the post-processed dataset reveals an underestimation of annual upwelling frequency  of ~1% (not shown) which is of the same magnitude order as the models error. Therefore it is important to note that in order to assess the ability of the regional model to simulate coastal upwelling, the choice of the satellite data set is crucial."

[Figure]

*Figure R1: Difference of annual upwelling frequency between the raw AVHRR dataset and the post-processed dataset. The difference in average and standard deviation are shown in the bottom-right corner.*

**Specifically, the upwellings detection method used in this study is that of Lehmann et al, 2012 despite the biais from the position of the coastlines whose axis is different from the East/West axis. Why this choice of method? Why not use another method as described in Schlegetk & Smit; 2018 and Abraham, Schlegetk & Smit; 2021.**

The method we chosed is easy to implement and has been tested and applied many times in the Baltic Sea (e.g. Lehmann et al., 2012; Gurova et al., 2013 Dutheil et al., 2021) contrary to the suggested methods. Nevertheless we acknowledge that the suggested method can indeed used to avoid the bias related to the orientation of the coast. However, in the original study of

Abrahams A, Schlegel RW, Smit AJ (2021) A novel approach to quantify metrics of upwelling intensity, frequency, and duration. PLoS ONE 16(7): e0254026. https://doi.org/10.1371/journal.pone.0254026

the suggested method was adapted to the coast of South Africa and requires the choice of certain thresholds and includes also the wind field evaluation. Hence, more investigation and intense analysis will be necessary to adapt this method to the Baltic Sea which is beyond the scope of this study. However, we are encouraged to to the work and adapt the method for the Baltic Sea in a followup study with specific focus on upwelling.

References

Gurova, E., Lehmann, A., Ivanov, A.: Upwelling dynamics in the Baltic Sea studied by a combined SAR/infrared satellite data and circulation model analysis, Oceanologia, 55(3), 687-707.DOI: 10.5697/oc.55-3.687

Dutheil, C., Meier, H.E.M., Gröger, M. and Boergel, Understanding past and future sea surface temperature trends in the Baltic Sea. Clim Dyn **58**, 3021–3039 (2022). https://doi.org/10.1007/s00382-021-06084-1

**Results:**

**In the introduction of the results, it is stated that different runoffs were for the HBM model.**
**This part should be in the material and method explaining the reason for this choice and**
**specifying which runoff were used.**

As HBM is an operational setup, it is straight forward for its implementation to utilize the respective runoff data set for this purpose. Nonetheless, the hydrological dataset is derived from the same source as for other models, i.e. E-HYPE forecasts (Donnelly et al., 2016).

Donnelly, C., Andersson, J.C., Arheimer, B., 2016. Using flow signatures and catchment similarities to evaluate the E-HYPE multi-basin model across Europe. Hydrol. Sci. J. 61 (2), 255–273. http://dx.doi.org/10.1080/02626667.2015.1027710.

We will include this note in the methods section of the revised version.

**In the first part of the results the role of thermoclyne formation in the sensitivity of the SST**
**to variations in meteorological forcing is stated but sparsely discussed. This lacks discussion**
**and bibliographic references.**

We agree. We will add a short paragraph on the role of thermocline formation and add bibliographic references.

**The section dealing with seasonality needs to be restructured. Suggestion: Discuss the**
**divergences of the models, station by station, with respect to temperature and then do the**
**same for salinity, in the same way as the introduction to Figure 5. Indeed, the paragraphs**

introducing the stations describe sometimes the variability of temperature, sometimes that of salinity.

**The discussion of temperature variability for the Nemo model is missing.**

Thank you for the suggestion. We will think about the structure and revise it accordingly. We will also include NEMO temperature variability in the discussion.

**Long term variability: In this part we still refer to the stability of the models. It is therefore**

**necessary to put the figures that illustrate these remarks in the publication.**

Yes, the section 3.4 "Long term variability of temperature and salinity" shows deep water time series which are related also to stability. The long-term development of salinity is a good indicator for this. The salinity at the deep stations BY15 and F9 show that for all models but HBM there are no significant drifts. We will include a remark about this in the revised version. However, we want to avoid any further analysis and production of new figures on this issue as this is beyond the scope of the manuscript.

**Also, in this and several times, it is referred to divergences of models because of their different**

**management of ice modules, what about turbidity that can limit the heat flux?**

It is true water turbidity and of course the individual models' light penetration scheme will also influence the heat fluxes in addition to sea ice. We will make a remark about this in the revised manuscript.

**Marine heat waves: Figure 8 with Table 1 again confirm what was explained in section 3.5**

**without adding additional information. It would be interesting to compare the models with the**

**data in Figure 8 to see which model is closer to the observed extreme values and not just that**

**the models diverge more for extreme temperatures.**

Figure 8 shows the annual mean surface and bottom temperatures averaged over the entire Baltic Sea. Table 1 is an overview showing model setup characteristics.

Maybe you mean Table 2 which lists yearly mean and maximum surface and bottom temperature trends in the spatial averages over the Baltic Sea? We agree a comparison with observed extreme values would be very interesting but to our knowledge no observational data sets exist that would allow the calculation of such long term trends in spatial averages over the entire Baltic Sea and over such a long time. Thus, this would require additional intense processing of observational data to allow a reasonable comparison with the models. This work is however, beyond the scope of our study which aims to highlight model differences (and thus uncertainty) despite one and the same forcing.

**Upwelling: In figure 11, the GMT_1nm model is analyzed, while in figure 12, GMT_2nm is**

**analyzed. Why this choice and why not treat the outputs of the MOM_1nm model in the**

**upwellings analysis?**

We agree. Due to the delays in the production of the simulations there was an offset between analysis and data availability from the respective models. However, meanwhile all analysis is complete and we will include MOM_1nm in the upwelling analysis.

**Water column stratification: This section ends with "Further detailed analyses of model**

**output may reveal the reasons underlying the difference in the timing of thermocline formation**

**despite identical atmospheric forcing." What do you suggest? This section should be discussed with references.**

Thank you for the comment. We will include a short note on what could be investigated in further studies to elaborate on the timing of thermocline formation, such as vertical turbulence schemes, the momentum transfer from wind into the sea, or different schemes for the light penetration into the water column. We will also include references for this.

**Summary :**

**In the conclusion, taking Hordoir et al., 2019 as an example of non-validated models in long-term simulations is not accurate because, in the first instance, the HBM model was chosen in the experiments as an example of an operational model. Furthermore, in Hordoir et al. 2019, the model is described as one that allows research on long-term simulations as much as on operational applications and whose simulations are devoid of data assimilation.**

We agree. The Hordoir et al., 2019 model is validated for long term multi-decadal simulations. We will remove Hordoir et al., 2019 in this context and refer solely to HBM. Thank you for the correction.

**Finally, salinity has once again been little discussed even though it is strongly impacted by runoffs, MBI…**

That's true. This reflects also that salinity dynamics is very complex in the Baltic Sea. In this first BMIP introduction paper, however, we can not go to deep into the details. Definitely this interesting topic will be taken up in follow-up studies.

**Technical corrections**

We thank the reviewer for the technical suggestions given below to improve the figures. We will revise the figures accordingly to facilitate the interpretation for the readers. We also thank for the correction of the reference list.

Reference error. This is not an exhaustive list

Name written differently:

Meier HEM, Döscher R, Coward AC, Nycander J, DöösK: RCO—Rossby Centre regional Ocean climate model: model description (version 1.0) and first results from the hindcast period 1992/93. Reports Oceanography No. 26, SMHI, Norrköping, Sweden, p 102, 1999.

Meier, H. E. M., and S. Saraiva : Projected Oceanographical Changes in the Baltic Sea until 2100. Oxford Research Encyclopedia of Climate Science, online publication date:. DOI: 10.1093/acrefore/9780190228620.013.69, 2020.

Meier, H.E.M., Dieterich, C., Gröger, M.: Natural variability is a large source of uncertainty in future projections of hypoxia in the Baltic Sea. Commun Earth Environ 2, 50 (2021). https://www.nature.com/articles/s43247-021-00115-9, 2021a.

Listed as duplicates:

Meier, H. M., Höglund, A., Döscher, R., Andersson, H., Löptien, U., & Kjellström, E. (2011). Quality assessment of atmospheric surface fields over the Baltic Sea from an ensemble of regional climate model simulations with respect to ocean dynamics. Oceanologia, 53, 193-227

**Figures**

**Fig.3 Use a different color palette for absolute values and differences for better readability.**

**Figure 3.e does not seem to have a colorbar with such a layout. Correct the extends of the colorbars that look truncated.**

**Fig.5 :Negative temperatures referred to in the text are not displayed on the scale**

**-Fig.5.a put the colorbar at the end of the figure horizontally**

**-Use the same width for all colorbars**

**-Center the station names-**

**Fig.5.b set the colorbar below each figure concerned and horizontally**

**Fig.10 : Reorganize the colorbars, the choice of palettes is not appropriate, the**

**Fig10.c and Fig.10.d seem to have the same color palette**

---

## Author Comment (AC2)

**Reply to reviewer 2**

**"The Baltic Sea model inter-comparison project BMIP - a platform for model development, evaluation, and uncertainty assessment" by Gröger et al. 2022**
**The manuscript presents a MIP for Baltic Sea models, which to this date has not existed before. So far, 4 models are participating, but 2 come in different resolutions which gives 6 in total. There is also a 7th model but the data availability from this model seems to be very limited as it is not presented in most plots. The models are forced by the same surface fluxes (atm. forcing and river input) and this surface forcing is presented as well. All models are compared to available observations and reanalysis products and some striking differences are found.**
**I think the paper is overall very interesting and well written. I would recommend it for publication after some minor comments below are addressed to make it more readable.**

We thank the reviewer for his thorough reading of the manuscript and the specific comments which we will consider and which will help to improve the manuscript significantly.

**Overall:**

**The authors rarely use the word "bias" in when discussing the differences between obs and model results. For example, line 309 says "positive anomalies in comparison with BSH climatology", but this occurs several other times in the manuscript. I would use the word "bias" more often to make the text easier to read.**

We agree and will use the word "bias" to replace the other terms.

**The resolution and/or size of almost all figures (3,5,6,11,14 seem the worst) is pretty poor. Perhaps the final layout will be different, but I struggled to even find the results in some figures when on printed paper. Model names are hard to read in Fig 5, and the lack of a coastline or filled continents in Fig 11 is odd.**
**I would recommend making the figures larger (maybe taking up a full page), and that the authors add coastlines or filled land in Fig 11.**

We agree figures have to be improved and we will definitely revise them (also in accordance with suggestions of rev#1). By it's nature, model comparisons with many models require much place and so individual maps are somehow constrained in size. We have however ensured a quality of the pictures of 300 dpi resolution so that at least on a screen individual maps can be adequately assessed.

**Detailed comments:**
**Page 3, Line 75: The additional reference to Myrberg 2010 is superfluous in my mind since it was given already in the previous sentence.**
We agree and will remove the reference.

**Page 5, line 158: I am very curious to know more details about the atmospheric forcing that is used here. I'm not familiar with UERRA. The website listed is not actually a set of instructions, but instead just a website for the project. Also, I would prefer if the authors spent some time in the main text of the manuscript to describe the data rather than point the reader to an external website.¨The authors should describe:**

**1) What the shortcomings are and what the corrections are.**
**2) What radiation was used? 2M temperature etc can be taken from analysis fields, but radiation and other fluxes must come from forecasts. In my experience, one would typically use the difference between the +6 and +12h forecasts of radiation, but I'd like to know what the authors do here.**
**3) Are any corrections needed for radiation? The commonly used DRAKKAR forcing set 5.2 (https://www.drakkar-ocean.eu/publications/reports/report_DFS5v3_April2016.pdf) had to do quite some corrections to the radiation fields. 4) Is there any effort in the data set to ensure that the surface water budget is closed, i.e. E-P-R = 0 over some time scale, or is**

**this done by the individual models?**

General reply:
We appreciate that the reviewer shows particular interest in the applied forcing data and we are happy to share more information on it. In the article, several websites are listed that are related to the forcing data, e.g.:
- line 143/144: a link to homepage of the service
- line 155: a link to a GitHub page is given, which includes source code explaining how UERRA-HARMONIE data can be prepared for NEMO-Nordic. When following the link from line 155, please choose "create_forcing_for_NEMO". There, you will find Python and shell scripts that were used to prepare the forcing data.
In case the reviewer is interested to learn more about the most recent regional reanalysis for Europe, the reviewer might take a look at CERRA (Copernicus European Regional ReAnalysis). CERRA was released in August 2022 and has a horizontal resolution of 5.5km and the same domain as UERRA-HARMONIE. Data are available in the CDS, here:
https://cds.climate.copernicus.eu/cdsapp#!/dataset/reanalysis-cerra-single-levels?tab=overview

Specific replies:

Reply 1)
Major shortcomings of the dataset are related to parameters based on the forecast model only. Hence parameters, which are not assimilated. A prominent example here is the total precipitation. The total precipitation is overestimated in the UERRA-HARMONIE dataset and therefore it is reduced by 20% for BMIP. The UERRA-HARMONIE cloudiness was corrupted in the post-processing step before archiving. Unfortunately, a cloud cover of 100% was archived as cloud free (0%). Therefore, it is suggested to use coastDat-2 cloudiness in the BMIP-context. Otherwise, no corrections were made to the UERRA-HARMONIE data. This information is included in the Suppl. Mat. S6. However, we will give this information also in the main document in a revised version.

Reply 2)
Analyses are only available at 00 UTC, 06 UTC, 12 UTC and 18 UTC but the forcing frequency is hourly. Hence, data at time stamps without analyses are from the forecast model. For analyzed parameters (e.g. temperature) the forcing data is a blended set of analyses and forecasts as follows:
00 UTC (analysis), 01 UTC (forecast), 02 UTC (fc), 03 UTC (fc), 04 UTC (fc), 05 UTC (fc), 06 UTC (an), 07 UTC (fc), …
Parameters, which are not analyzed, are taken from the forecast model only.
To avoid spin-up effects after the data assimilation/analyses, longer forecasts can be used as mentioned by the reviewer. For the BMIP forcing, that is also done for precipitation. Here, we subtract the 12h forecast from the 24h forecast to avoid the model spin-up. For radiation, the BMIP forcing uses hourly data from the forecast model. The parameters "Time-intergrated surface solar radiation downwards" and "Time-intergrated surface thermal radiation downwards" were used.

Reply 3)
No correction was applied to the radiation parameters.

Reply 4)
The surface water budget in UERRA-HARMONIE is not closed. As explained above, the model precipitation is reduced by 20% and the runoff is based on observations.

**Page 6, line 171: "The GETM_1nm and GETM_2nm domain is limited to the southern Kattegat" this makes it sound like the model domain only covers the Kattegat, which I'm sure it does not. The sentence should rather be "The GETM_1nm and GETM_2nm domains cover the Baltic Sea including the Kattegat while the two MOM domains also include parts of the Skagerrak. Both the NEMO and HBM domains encompass the Baltic and the North Sea, for which they also use tidal forcing on the lateral boundary condition."**

Thank you very much for this correction which we will include in the revised manuscript.

**Page 8, line 256: "it is limited to regions where the coastline is mainly oriented along an east/west axis as in the Gulf of Finland". Does this mean the method is only applicable there? I think maybe you mean that the method is most applicable when the coastline is north/south, and not so well applicable where it's east/west?**

We agree that our formulation is ambiguous. This method calculates the difference with the zonal mean, so the method is most applicable when the coastline is oriented north/south.

We will modify this sentence as follows to avoid this confusion: "This method is less reliable in regions where the coastline is mainly oriented along an east/west axis as in the Gulf of Finland".

**The authors discuss the biases in upwelling along the Swedish coast (mainly meridional) and the Gulf of Riga (zonal and meridional) so if the method is less reliable for a specific direction, it could explain some of the larger biases they find.**

We agree with the reviewer and this is clearly a limitation of this method. Nevertheless, as mentioned, upwelling occurs mainly along the southern Swedish coast where this limitation does not occur. For the Gulf of Riga, this could be an explanation.

We will add this sentence to discuss this point: "Along the zonal coasts, we are not able to disentangle whether the bias is due to the model or to the limitation of the upwelling detection method.

**Fig 3. Perhaps the figure could be made to take up a full A4 page. It is very small and labels are difficult to read.**

We agree and will try make figure 3 to fill a full A4 page.

**Fig 5: This figure would also benefit from being larger.**

Will be done.

**Page 14, line 380: "NEMO_2nm showed that salinities were lowest in the deep later but highest in the upper layers". This makes it sound like NEMO and GETM simulate fresh bottom and salty upper ocean, which is surely not the case. I think the authors mean to say that NEMO and GETM are fresher at depth and saltier in the upper ocean compared to the other models.**

Thank you for the correction! That is indeed what we meant. We will correct the sentence in the revised version.

**Fig 6: This figure needs to be made larger. It is at times difficult to see the differences authors are referring to in the text.**

We agree and will make the figure larger.

**Fig 9: Please make the model names larger (can hardly be read when on printed A4 paper). Also please add a vertical line in the top-left subfigure to indicate in what year the reanalysis ends, and please explain this in the figure caption as well.**

Thank you. We will change Fig. 9 and the caption accordingly.

**Fig 11: Why is MOM_1nm and GETM_2nm not in this plot? Was the data not available? I think the authors computed the upwelling themselves using the temperature, i.e. it is not an online diagnostic, so it should be possible to do for both MOM and GETM as well. Or do those model runs, which differ only in resolution from their twins, produce the same result? I would think upwelling can be sensitive to the horizontal resolution. Also, I would strongly recommend adding filled land or coastlines in this plot to make it easier to view.**

Due to the delays in the production of the simulations there was an offset between analysis and data availability from the respective models. However, meanwhile all analysis is complete and we will complete the analysis. Hence, MOM_1nm and GETM_2nm will be included in the revised version. Also we will include coastlines in this plot.

**Figs 11,12. It strikes me that NEMO simulates a very different pattern of biases, and much smaller biases in upwelling overall. I understand the authors do not want to deep dive into why this is, but I think some speculation on why NEMO is so different could be warranted. The final answer could be left for future work.**

We agree and will more emphasize the low bias of NEMO and hypothesize about the reason.

**Fig 13: Why is GETM_2nm not in this plot?**

Will be include in the revised version. See also our reply to Fig. 11.

---

## Author Response (AR1)

**Reply to reviewer 1**

**General comments**
**The paper addresses relevant scientific modeling issues using a never-before-used and undeniably rigorous protocol for comparing numerical models of the Baltic Sea dynamics. This method is useful in determining the discrepancies between the models because the same forcings were applied to each of the models, thus allowing the discrepancies between the models to be interpreted as being due to the model's own configuration or grid resolution. The authors have well introduced the lack of inter-comparison studies of regional models against other (global) models and the need to do so for the Baltic Sea and the North Sea. To justify the importance of this exercise, they highlighted the complexity of the site and the diversity of dynamic models used to simulate, among others, the general circulation. A state of the art of the models used in the study area is quite complete, it is presented at the beginning of the article. The detailed method is available on the project's website and allows for identical replication of the experiments. The added value is in the potential reproducibility of the method to other marginal seas. Thus, the technical approach is clearly explained with some exceptions that will be mentioned in the "Specific Comment" section. The overall structure of the paper and its presentation make it clear and easy to read. However, some parts of the results need revision which I detail in the "Specific Comment". In addition, many of the figures need to be reworked. However, the main messages of the publication are clear but the results lack discussion and consideration of related work.**

We gratefully thank the reviewer for his/her thorough reading and the specific comments which we have fully considered and which led to substantial improvements of the manuscript..

**Abstract**
**It is detailed and includes the main results. The absence of information about salinity is deplored.**

That's a good idea. We have added a short statement highlighting the main salinity results.

"The spread of water salinity across the models is generally larger compared to water temperature, which is most obvious in the long term time series of deep water salinity. The inflow dynamics of saline water from the North Sea is covered well by most models, but the magnitude as inferred from salinity differs much as well as the simulated mean salinity of deep waters." (line 35)

**Methods:**
**This part is very clear, which makes it easy to notice the few missing information.**

**Runoff used for this study are clearly referenced but the way they are implemented is not explained, if the runoff is added to the first mesh from the coast or diffused over several meshes, if they are applied only on the surface or on the whole water column?**

The implementation is not specified in the protocol because the models manage this differently, as it is constrained by internal model configuration such as the vertical and horizontal grid spacing and options in the model code.

We have added the following paragraph in section 2.2. Ocean models (lines 198ff):

"While the BMIP provides time series of runoff, the implementation may differ between models due to technical design and model setup (e.g. horizontal and vertical resolution). For this study runoff was implemented along the coasts as follows:

**HBM, MOM**: runoff is added to one coastal grid cell in the top layer
**GETM**: always in the top cell
discharge < 500 m³/s - one cell discharge near the coast.
discharge < 1000 m³/s - spread evenly over two horizontal cells near the coast
discharge > 1000 m³/s - spread evenly over three horizontal cells near the coast
**NEMO**: one cell discharge near the coast. Distributed over the whole water column. "

**Moreover, contrary to the choice of atmospheric forcing which is justified, the choice of runoff is not explained.**

As there are no homogeneous river discharge datasets for the entire period 1961-2018 available and because the last years were only covered by the E-HYPE model forecast product, we merged the two E-HYPE model datasets and the observational records. At least for the basin averages the BMIP dataset is homogeneous and consistent. We have added the following statement at the beginning of section 2.1 XXX – XXX Forcing – Runoff data (line 132)

"Since runoff is a key parameter for most brackish marginal seas, the availability of physically consistent multi-decadal discharge time series are a prerequisite for climate modeling in the Baltic Sea. As there are no homogeneous river discharge datasets for the entire period 1961-2018 available and because for the last years were only covered by an E-HYPE model forecast product, a new homogeneous runoff product was produced within the BMIP project"

**Spin-up : There are many references to model stability in the article, however, in the supplementary material there is no figure showing the stability of each of the models. Although the recommendations are clear, they are not explained. Why is it recommended to run the simulations again in July 2004 and not another month? What initialization was used to start the spin up runs?**

The BMIP protocoll provides no initial data for the start of the spinup. As the Baltic Sea has an overturning time of about 30 years, BMIP gives the conservative recommendation for a 1961-2004 (44 years, thus > than the Baltic Sea overturning time) spinup to reach an equlibrium where potential drifts can be minimized. BMIP recommends to start the production runs in mid-summer as in this season Major Baltic Sea salt inflows (MBI) from the North Sea are extremely unlikely.
We will make this more clear in the revised version. We have added the following statement in section 2. 3 The BMIP protocol version 1.0 (line 243):

"This calendar date represents calm climatic conditions around mid-summer and therefore avoids drastic changes at the beginning of the production runs. The duration of the recommended 44-year spinup is a compromise between the computational resources to drive high resolution models and the need to minimize model drifts. Therefore the spinup run should be significantly longer than the internal overturning period in the Baltic Sea which is estimated about 30 years (Placke et al., 2021). "

**Also implied by these comments is the issue of applying the same spin-up for all models despite differences in grid resolution and turbulence schemes.**

This is correct. Model internal turbulence schemes and resolution may also influence the time the model reaches equilibrium. That's why we recommend an at least 44 year spinup duration (>overturning time) which is a compromise between costs to drive the model and the minimization of potential drifts.

We have added the following statement in section 2. 3 The BMIP protocol version 1.0 (lines 249):

"Due to the diversity of present and potential future Baltic Sea models the spinup recommendations do not guarantee a perfect model equilibrium at the end of the spinup integration as the duration to reach that equilibrium will also depend on the specific model configuration such as model grid resolution, turbulence schemes, or bottom boundary layer formulations, which all may differ among the models."

**Analysis methods : It may be of interest to indicate the error associated with the postprocessing of AVHRR data.**

Thank you for your comment. To address it, we downloaded the raw AVHRR dataset and compared it with the post-processed dataset (Fig R1). This figure shows that the raw AHRR dataset underestimates the upwelling frequency by 0.9% but overestimates the spatial variability because it overestimates the frequency in Bothnian Bay. This result is consistent with the principle of post-processing as it unmasks regions misidentified by the cloud detection algorithm.

We added the following sentences in 2.4. Analysis of heat waves, coastal upwelling, and water column stratification (lines 289):

"A comparison between raw AVHRR dataset and the post-processed dataset reveals an underestimation of annual upwelling frequency of ~1% (not shown) which is of the same magnitude order as the models error. Therefore it is important to note

that in order to assess the ability of the regional model to simulate coastal upwelling, the choice of the satellite data set is crucial."

[Figure]

*Figure R1: Difference of annual upwelling frequency between the raw AVHRR dataset and the post-processed dataset. The difference in average and standard deviation are shown in the bottom-right corner.*

**Specifically, the upwellings detection method used in this study is that of Lehmann et al, 2012 despite the biais from the position of the coastlines whose axis is different from the East/West axis. Why this choice of method? Why not use another method as described in Schlegetk & Smit; 2018 and Abraham, Schlegetk & Smit; 2021.**

The method we chosed is easy to implement and has been tested and applied many times in the Baltic Sea (e.g. Lehmann et al., 2012; Gurova et al., 2013 Dutheil et al., 2021) contrary to the suggested methods. Nevertheless we acknowledge that the suggested method can indeed used to avoid the bias related to the orientation of the coast. However, in the original study of

Abrahams A, Schlegel RW, Smit AJ (2021) A novel approach to quantify metrics of upwelling intensity, frequency, and duration. PLoS ONE 16(7): e0254026. https://doi.org/10.1371/journal.pone.0254026

the suggested method was adapted to the coast of South Africa and requires the choice of certain thresholds and includes also the wind field evaluation. Hence, more investigation and intense analysis will be necessary to adapt this method to the Baltic Sea which is beyond the scope of this study. However, we are encouraged to to the work and adapt the method for the Baltic Sea in a followup study with specific focus on upwelling.

We added a sentence to the Methods section 2.4 / Upwelling

"Recently, more advanced methods to detect upwelling that circumvent the problem of coastline orientation were developed for other marginal seas (Abraham et al., 2021) that can be adapted to the Baltic Sea in future studies." (lines 301)

References
Gurova, E., Lehmann, A., Ivanov, A.: Upwelling dynamics in the Baltic Sea studied by a combined SAR/infrared satellite data and circulation model analysis, Oceanologia, 55(3), 687-707.DOI: 10.5697/oc.55-3.687

Dutheil, C., Meier, H.E.M., Gröger, M. and Boergel, Understanding past and future sea surface temperature trends in the Baltic Sea. Clim Dyn **58**, 3021–3039 (2022). https://doi.org/10.1007/s00382-021-06084-1

**Results:**
**In the introduction of the results, it is stated that different runoffs were for the HBM model.**

**This part should be in the material and method explaining the reason for this choice and specifying which runoff were used.**

We agree and provide the following paragraph in section 2.2 Methods/ ocean models section to make this more clear:

"As HBM is an operational setup, it is straight forward for its implementation to utilize the respective runoff data set for this purpose. Thus HBM differs from the other models with respect to the used data. Nonetheless, the hydrological dataset used for HBM in this study is derived from the same source as for other models, i.e. E-HYPE forecasts (Donnelly et al., 2016)." (lines 208).

**In the first part of the results the role of thermoclyne formation in the sensitivity of the SST to variations in meteorological forcing is stated but sparsely discussed. This lacks discussion and bibliographic references.**

We agree. We added a few lines highlighting the importance of the thermocline dynamics in the Baltic Sea in the paragraph discussing the thermocline importance and to refer to the relevant literature. The whole paragraph reads now (addition in green font):

During summer, meteorological forcing was characterized by calm winds and stronger solar radiation, which promoted an intense thermal layering of the uppermost water column. In the open sea, air-sea coupling was affected by the presence of a strong thermocline that reduced exchange with cooler waters from greater depths. The subsequent reduction in the effective water column heat capacity made the SSTs more prone to variations in meteorological forcing than was the case in winter. Hence, the seasonal and spatial characteristics of the thermocline as well as its intensity and its position in the water column is a key parameter for the development of of SSTs right from its formation in spring to its disappearance in autumn and is well investigated in the Baltic Sea (e.g.Eilola, 1997; Liblik and Lips, 2011; Hordoir and Meier, 2012; Chubarenko et al., 2017; Liblik and and Lips, 2019; Gröger et al., 2019).

**The section dealing with seasonality needs to be restructured. Suggestion: Discuss the divergences of the models, station by station, with respect to temperature and then do the same for salinity, in the same way as the introduction to Figure 5. Indeed, the paragraphs introducing the stations describe sometimes the variability of temperature, sometimes that of salinity. The discussion of temperature variability for the Nemo model is missing.**

Thank you for the suggestion. We agree. We have restructured section "3.3 Mean seasonal cycle". The discussion for water temperature and salinity are now separately discussed station by staion in section 3.3.1 Water temperature and 3.3. Salinity.

NEMO is now explicitly in the discussion of all station description with respect of salinity which show highest inter-model differences. It is also noted for temperature for station Anholt where NEMO has a too sallow bathymetry.

**Long term variability: In this part we still refer to the stability of the models. It is therefore necessary to put the figures that illustrate these remarks in the publication.**

Yes, the section 3.4 "Long term variability of temperature and salinity" shows deep water time series which are related also to stability. The long-term development of salinity is a good indicator for this. The salinity at the deep stations BY15 and F9 show that for all models but HBM, that there are no significant drifts.

However, we want to avoid any further analysis and production of new figures on this specific issue as this is beyond the scope of the manuscript.

We added the following sentence at the end of section 3.4 "Long term variability of temperature and salinity" to make ths clear lines (492):

"All models but the operational setup HBM, show no significant drift in the deep salinities at stations F9 and BY15 and so confirm the length of the spinup run of 44 years."

**Also, in this and several times, it is referred to divergences of models because of their different management of ice modules, what about turbidity that can limit the heat flux?**

It is true water turbidity and of course the individual models' light penetration scheme will also influence the heat fluxes in addition to sea ice. We will made the following remark about this in the revised manuscript.

"Beside sea ice, other processes influencing the air sea heatfluxes like differences in water turbidity or the models optical light penetration schemes should be considered in this context as well." (line 705)

**Marine heat waves: Figure 8 with Table 1 again confirm what was explained in section 3.5**

**without adding additional information. It would be interesting to compare the models with the data in Figure 8 to see which model is closer to the observed extreme values and not just that the models diverge more for extreme temperatures.**

Figure 8 (now figure 9) shows the models' annual mean surface and bottom temperatures averaged over the entire Baltic Sea. Table 1 is an overview showing model setup characteristics.
Maybe you mean Table 2 which lists yearly mean and maximum surface and bottom temperature trends in the spatial averages over the Baltic Sea? We agree a comparison with observed extreme values would be very interesting but to our knowledge no observational data sets exist that would allow the calculation of such long term trends in spatial averages over the entire Baltic Sea and over such a long time. Thus, this would require additional intense processing of observational data to allow a reasonable comparison with the models. This work is however, beyond the scope of our study which aims to highlight model differences (and thus uncertainty) despite one and the same forcing.

**Upwelling: In figure 11, the GMT_1nm model is analyzed, while in figure 12, GMT_2nm is analyzed. Why this choice and why not treat the outputs of the MOM_1nm model in the upwellings analysis?**

This figure is now figure 12. Due to the delays in the production of the simulations there was an offset between analysis and data availability from the respective models. MOM_1nm and GETM_2nm are now included in the revised version. Also the coastlines are now clearly visible.
With this, MOM and GETM are included in both, their high and low resolution versions. The high resolution versions show the same pattern as their low resolution twins but with a higher bias. We added a short remark about (line 586).

"The high resolution versions MOM_1nm and GETM_1nm (Fig. 12d, and g) show in most regions a larger bias to the observations than their coarse resolution versions MOM_3nm and GETM_2nm."

**Water column stratification: This section ends with "Further detailed analyses of model output may reveal the reasons underlying the difference in the timing of thermocline formation despite identical atmospheric forcing." What do you suggest? This section should be discussed with references.**

Thank you for the comment. We have includes a short note on what could be investigated in further studies to elaborate on the timing of thermocline formation, with citing relevant literature in this context include references for this.

The sentence now reads (line 645):
"Further detailed analyses of model output may reveal the reasons underlying the difference in the timing of thermocline formation despite identical atmospheric forcing which may involve the models formulations for turbulence, the effect of grid resolution or bottom friction (e.g. Krauss, 1981; Eilola, 1997; Liblik and Lips, 2011; Hordoir and Meier, 2012; Chubarenko et al., 2017; Liblik and and Lips, 2019; Gröger et al., 2019)."

However, we want to avoid vast speculations about thermocline dynamics in the models without further analysis.

**Summary :**
**In the conclusion, taking Hordoir et al., 2019 as an example of non-validated models in long-term simulations is not accurate because, in the first instance, the HBM model was chosen in the experiments as an example of an operational model. Furthermore, in Hordoir et al. 2019, the model is described as one that allows research on long-term simulations as much as on operational applications and whose simulations are devoid of data assimilation.**

We agree. The Hordoir et al., 2019 model is validated for long term multi-decadal simulations. We have removed Hordoir et al., 2019 in this context. Thank you for the correction.

**Finally, salinity has once again been little discussed even though it is strongly impacted by runoffs, MBI…**
That's true. This reflects also that salinity dynamics is very complex in the Baltic Sea. In this first BMIP introduction paper, however, we can not go to deep into the details. Definitely this interesting topic will be taken up in follow-up studies.

**Technical corrections**

We thank the reviewer for the technical corrections and suggestions given below to improve the figures. We will revise the figures accordingly to facilitate the interpretation for the readers. We also thank for the correction of the reference list.

**Reference error. This is not an exhaustive list**

**Name written differently:**

**Meier HEM, Döscher R, Coward AC, Nycander J, DöösK: RCO—Rossby Centre regional Ocean climate model: model description (version 1.0) and first results from the hindcast period 1992/93. Reports Oceanography No. 26, SMHI, Norrköping, Sweden, p 102, 1999.**

**Meier, H. E. M., and S. Saraiva : Projected Oceanographical Changes in the Baltic Sea until 2100. Oxford Research Encyclopedia of Climate Science, online publication date:. DOI: 10.1093/acrefore/9780190228620.013.69, 2020.**

**Meier, H.E.M., Dieterich, C., Gröger, M.: Natural variability is a large source of uncertainty in future projections of hypoxia in the Baltic Sea. Commun Earth Environ 2, 50 (2021). https://www.nature.com/articles/s43247-021-00115-9, 2021a.**

We have put the reference in the right order now and corrected the typos. The double entry was removed.

**Listed as duplicates:**
**Meier, H. M., Höglund, A., Döscher, R., Andersson, H., Löptien, U., & Kjellström, E. (2011). Quality assessment of atmospheric surface fields over the Baltic Sea from an ensemble of regional climate model simulations with respect to ocean dynamics. Oceanologia, 53, 193-227**

Thank you. The duplicate entry was removed now.

**Figures**

**Fig.3 Use a different color palette for absolute values and differences for better readability.**
**Figure 3.e does not seem to have a colorbar with such a layout. Correct the extends of the colorbars that look truncated.**

We have completely revised figure 3 and split it into two figures. Absolute and differences plots have now a different well-distinguishable color palettes. Figure 3 displays now climatologies for summer and winter. The previously incorporated correlation maps are now put into the new figure 4. Thus, both figures are substantially larger now and better readable.

**Fig.5 :Negative temperatures referred to in the text are not displayed on the scale**
**-Fig.5.a put the colorbar at the end of the figure horizontally**
**-Use the same width for all colorbars**
**-Center the station names-**
**Fig.5.b set the colorbar below each figure concerned and horizontally**

This figure turned to figure 6 in the revised paper. Its now larger and the color bars are of even length and were placed below each panel and so are better visible.

**Fig.10 : Reorganize the colorbars, the choice of palettes is not appropriate, the**
**Fig10.c and Fig.10.d seem to have the same color palette**
This is now Figure 11. We have reorganized the color bars. The color palettes were intended in that way. Figure a) and b) show absolute climatologies MHW frequency and duration, while c) and d) show their relative change from 1965-1989 to 1994-2018. Figs. 10c) and d) have the same color palette but show a different scaling. We made a note on that in the caption.

**Reply to reviewer 2**

**"The Baltic Sea model inter-comparison project BMIP - a platform for model development, evaluation, and uncertainty assessment" by Gröger et al. 2022 The manuscript presents a MIP for Baltic Sea models, which to this date has not existed before. So far, 4 models are participating, but 2 come in different resolutions which gives 6 in total. There is also a 7th model but the data availability from this model seems to be very limited as it is not presented in most plots. The models are forced by the same surface fluxes (atm. forcing and river input) and this surface forcing is presented as well. All models are compared to available observations and reanalysis products and some striking differences are found. I think the paper is overall very interesting and well written. I would recommend it for publication after some minor comments below are addressed to make it more readable.**

We thank the reviewer for his thorough reading of the manuscript and the specific comments. We have accounted the all the suggestions which led to a significant improvement of the manuscript.

**Overall:**

**The authors rarely use the word "bias" in when discussing the differences between obs and model results. For example, line 309 says "positive anomalies in comparison with BSH climatology", but this occurs several other times in the manuscript. I would use the word "bias" more often to make the text easier to read.**

We agree. In the revised version have replaced the terms "deviation", "anomaly", and difference by the word "bias" when models are compared to observations. Mismatches between the models were referred as "differences".

**The resolution and/or size of almost all figures (3,5,6,11,14 seem the worst) is pretty poor. Perhaps the final layout will be different, but I struggled to even find the results in some figures when on printed paper. Model names are hard to read in Fig 5, and the lack of a coastline or filled continents in Fig 11 is odd. I would recommend making the figures larger (maybe taking up a full page), and that the authors add coastlines or filled land in Fig 11.**

We agree. We have completely revised the figure by restructuring figure elements, changing color bars. Figures were enlarged as recommended by bot reviewers and text elements were increased in size, coastlines were added etc.. See detailed description in the pointwise replies below.

**Detailed comments:**
**Page 3, Line 75: The additional reference to Myrberg 2010 is superfluous in my mind since it was given already in the previous sentence.**
We agree and have removed the "double"-reference.

**Page 5, line 158: I am very curious to know more details about the atmospheric forcing that is used here. I'm not familiar with UERRA. The website listed is not actually a set of instructions, but instead just a website for the project. Also, I would prefer if the authors spent some time in the main text of the manuscript to describe the data rather than point the reader to an external website. The authors should describe:**

**1) What the shortcomings are and what the corrections are.**
**2) What radiation was used? 2M temperature etc can be taken from analysis fields, but radiation and other fluxes must come from forecasts. In my experience, one would typically use the difference between the +6 and +12h forecasts of radiation, but I'd like to know what the authors do here.**
**3) Are any corrections needed for radiation? The commonly used DRAKKAR forcing set 5.2 (https://www.drakkar-ocean.eu/publications/reports/report_DFS5v3_April2016.pdf) had to do quite some corrections to the radiation fields. 4) Is there any effort in the data set to ensure that the surface water budget is closed, i.e. E-P-R = 0 over some time scale, or is this done by the individual models?**

General reply:
We appreciate that the reviewer shows particular interest in the applied forcing data and we are happy to share more information on it. In the article, several websites are listed that are related to the forcing data, e.g.:
- line 143/144: a link to homepage of the service
- line 155: a link to a GitHub page is given, which includes source code explaining how UERRA-HARMONIE data can be prepared for NEMO-Nordic. When following the link from line 155, please choose "create_forcing_for_NEMO". There,

you will find Python and shell scripts that were used to prepare the forcing data.

In case the reviewer is interested to learn more about the most recent regional reanalysis for Europe, the reviewer might take a look at CERRA (Copernicus European Regional ReAnalysis). CERRA was released in August 2022 and has a horizontal resolution of 5.5km and the same domain as UERRA-HARMONIE. Data are available in the CDS, here: https://cds.climate.copernicus.eu/cdsapp#!/dataset/reanalysis-cerra-single-levels?tab=overview

Specific replies:

Reply 1)
Potential shortcomings of the dataset are related to parameters based on the forecast model only. Hence parameters, which are not assimilated. A prominent example here is the total precipitation. The total precipitation is overestimated in the UERRA-HARMONIE dataset and therefore it is reduced by 20% for BMIP. The UERRA-HARMONIE cloudiness was corrupted in the post-processing step before archiving. Unfortunately, a cloud cover of 100% was archived as cloud free (0%). Therefore, it is suggested to use coastDat-2 cloudiness in the BMIP-context. Otherwise, no corrections were made to the UERRA-HARMONIE data. This information is included in the Suppl. Mat. S6. However, we will give this information also in the main document in a revised version.

Reply 2)
Analyses are only available at 00 UTC, 06 UTC, 12 UTC and 18 UTC but the forcing frequency is hourly. Hence, data at time stamps without analyses are from the forecast model. For analyzed parameters (e.g. temperature) the forcing data is a blended set of analyses and forecasts as follows:
00 UTC (analysis), 01 UTC (forecast), 02 UTC (fc), 03 UTC (fc), 04 UTC (fc), 05 UTC (fc), 06 UTC (an), 07 UTC (fc), …
Parameters, which are not analyzed, are taken from the forecast model only.
To avoid spin-up effects after the data assimilation/analyses, longer forecasts can be used as mentioned by the reviewer. For the BMIP forcing, that is also done for precipitation. Here, we subtract the 12h forecast from the 24h forecast to avoid the model spin-up. For radiation, the BMIP forcing uses hourly data from the forecast model. The parameters "Time-integrated surface solar radiation downwards" and "Time-integrated surface thermal radiation downwards" were used.

Reply 3)
No correction was applied to the radiation parameters.

Reply 4)
The surface water budget in UERRA-HARMONIE is not closed. As explained above (and described in the Suppl. Mat, S6), the model precipitation must be reduced by 20% and the runoff is based on observations.

**Page 6, line 171: "The GETM_1nm and GETM_2nm domain is limited to the southern Kattegat" this makes it sound like the model domain only covers the Kattegat, which I'm sure it does not. The sentence should rather be "The GETM_1nm and GETM_2nm domains cover the Baltic Sea including the Kattegat while the two MOM domains also include parts of the Skagerrak. Both the NEMO and HBM domains encompass the Baltic and the North Sea, for which they also use tidal forcing on the lateral boundary condition."**

Thank you very much for this correction. We have changed the sentences accordingly (line 186).

**Page 8, line 256: "it is limited to regions where the coastline is mainly oriented along an east/west axis as in the Gulf of Finland". Does this mean the method is only applicable there? I think maybe you mean that the method is most applicable when the coastline is north/south, and not so well applicable where it's east/west?**

We agree that our formulation is ambiguous. This method calculates the difference with the zonal mean, so the method is most applicable when the coastline is oriented north/south.

We have modified this sentence to avoid this confusion as follows: " As this method is based on a difference with the zonal mean, it is less reliable in regions where the coastline is mainly oriented along an east/west axis as in the Gulf of Finland." (line 299).

**The authors discuss the biases in upwelling along the Swedish coast (mainly meridional) and the Gulf of Riga (zonal and meridional) so if the method is less reliable for a specific direction, it could explain some of the larger biases they find.**

We agree with the reviewer and this is clearly a limitation of this method. Nevertheless, as mentioned, upwelling occurs mainly along the southern Swedish coast where this limitation does not occur. For the Gulf of Riga, this could be an explanation.

We have added the following sentence to make this clear (line 572): "Along the zonal coasts, we are not able to disentangle whether the bias is due to the model or to the limitation of the upwelling detection method.

**Fig 3. Perhaps the figure could be made to take up a full A4 page. It is very small and labels are difficult to read.**
We have completely revised figure 3 and split it into two figures. Absolute and differences plots have now a different well-distinguishable color palettes. Figure 3 displays now climatologies for summer and winter. The previously incorporated correlation maps are now put into the new figure 4. Thus, both figures are substantially larger now and better readable.

**Fig 5: This figure would also benefit from being larger.**
Note this figure turned to figure 6 in the revised paper. Its now larger and the color bars are of even length and were placed below each panel and so are better visible.

**Page 14, line 380: "NEMO_2nm showed that salinities were lowest in the deep later but highest in the upper layers". This makes it sound like NEMO and GETM simulate fresh bottom and salty upper ocean, which is surely not the case. I think the authors mean to say that NEMO and GETM are fresher at depth and saltier in the upper ocean compared to the other models.**

Thank you for the correction! That is indeed what we meant. We rephrased the sentence as follows:

"Apart from HBM which was not designed to focus on MBIs, the GETM_2nm and NEMO_2nm are the models with lowest salinity in the deep layer but highest salinity in the upper layers of all models, suggesting stronger vertical mixing." (line 450).

**Fig 6: This figure needs to be made larger. It is at times difficult to see the differences authors are referring to in the text.**
This figure is now figure 7. The panels have been restructured into 4 panels a) -d). It was likewise made larger extending over almost 1 page.

**Fig 9: Please make the model names larger (can hardly be read when on printed A4 paper). Also please add a vertical line in the top-left subfigure to indicate in what year the reanalysis ends, and please explain this in the figure caption as well.**
Figure 9 is now Figure 10. Model names have been enlarged by 30% and vertical line now indicates the end of the reanalysis period in each panel.

**Fig 11: Why is MOM_1nm and GETM_2nm not in this plot? Was the data not available? I think the authors computed the upwelling themselves using the temperature, i.e. it is not an online diagnostic, so it should be possible to do for both MOM and GETM as well. Or do those model runs, which differ only in resolution from their twins, produce the same result? I would think upwelling can be sensitive to the horizontal resolution. Also, I would strongly recommend adding filled land or coastlines in this plot to make it easier to view.**

This figure is now figure 12. Due to the delays in the production of the simulations there was an offset between analysis and data availability from the respective models.  MOM_1nm and GETM_2nm are now included in the revised version. Also the coastlines are now clearly visible.
With this, MOM and GETM are included in both, their high and low resolution versions. The high resolution versions show the same pattern as their low resolution twins but with a higher bias. We added a  short remark about (line 586):

"The high resolution versions MOM_1nm and GETM_1nm (Fig. 12d, and g) show in most regions a larger bias to the

observations than their coarse resolution versions MOM_3nm and GETM_2nm (Fig. 12 a, and f)."

**Figs 11,12. It strikes me that NEMO simulates a very different pattern of biases, and much smaller biases in upwelling overall. I understand the authors do not want to deep dive into why this is, but I think some speculation on why NEMO is so different could be warranted. The final answer could be left for future work.**

We agree. In order to  emphasize the low bias of NEMO and to hypothesize about the reason we have included the following sentence (line 581).

"The opposite spatial bias pattern was determined for NEMO_2nm which shows likewise a very small bias overall. From the above descriptive statistics alone it is difficult  speculate about the different bias pattern seen in NEMO_2nm in particular in the Gulf of Riga. However, possible reasons, could include all modell formulations influencing variability such as atmosphere to ocean momentum transfer, seabed – water boundary layer. "

**Fig 13: Why is GETM_2nm not in this plot?**

Figure 13 is now figure 14. GETM_2nm is now included. As expected it shows the same pattern as the GETM_1nm version.

---

## Author Response (AR2)

**Comments to the author**:
**Dear author,**
**Thank you very much for this revised manuscript that answers in detail all reviewer's comment .**
**I just would like you to consider the following additional comments from my side :**

**- On p.6, you mention that «the hydrological dataset used for HBM in this study is derived from the same sources as for other models » but you also mention just before that the runoffs are different. Can you specify more precisely what is the nature of the hydrological datasets, besides runoffs, that are the same for all models?**

Thank you for pointing to this. This was missphrased (only the runoff data differs!):
We changed

"As HBM is an operational setup, it is straight forward for its implementation to utilize the respective runoff data set for this purpose. Thus HBM differs from the other models with respect to the used data. Nonetheless, the hydrological dataset used for HBM in this study is derived from the same source as for other models, i.e. E-HYPE forecasts (Donnelly et al., 2016)."

to (corrections in red)

As HBM is an operational setup, it is straight forward for its implementation to utilize the respective runoff data set for this purpose. Thus HBM differs from the other models with respect to the used runoff data. Nonetheless, the hydrological dataset used for HBM in this study is derived from the same source as for other models, i.e. E-HYPE forecasts (Donnelly et al., 2016). Nonetheless, the hydrological dataset used for HBM in this study is derived from the same source as for other models, i.e. E-HYPE forecasts (Donnelly et al., 2016), but did not undergo the harmonization procedure during the compilation of the official BMIP runoff data set. All other forcing data are exactly the same as for the other models.

**- On p.12, you discuss the «development of SSTs»; I don't really understand what «development of SSTs» mean. SSTs do not develop, they are always present with higher or lower values. Do you mean the development of SSTs extrema, or something similar? Or maybe, you should replace «development» by «evolution»?**

We completely agree and replaced "development" by "evolution"

**- Regarding the 2nd comment of the 2nd reviewer, please consider adding in the manuscript some details you present here but not in the text, e.g. your « Reply 2), 3) and 4).**

We have now included the information now in the main text on page 12 (lines 157-159; line 167)

Minor comments :

- On p.20, remove the comma after « the operational setup HBM »
- On p.27, l.585, add a space in « theseabed »
- On p.27, l.585, change « The, overall, ... » for « Thus, overall, ... »

Thank you. We adopted the corrections in the text.